# STING enhances cell death through regulation of reactive oxygen species and DNA damage

Thomas J. Hayman [1], Marta Baro[1], Tyler MacNeil[2], Chatchai Phoomak [1], Thazin Nwe Aung[2], Wei Cui[1], Kevin Leach[3], Radhakrishnan Iyer[3], Sreerupa Challa[3], Teresa Sandoval-Schaefer[4], Barbara A. Burtness[5], David L. Rimm [2] & Joseph N. Contessa [1,6 ✉]

Resistance to DNA-damaging agents is a significant cause of treatment failure and poor outcomes in oncology. To identify unrecognized regulators of cell survival we performed a whole-genome CRISPR-Cas9 screen using treatment with ionizing radiation as a selective pressure, and identified STING (stimulator of interferon genes) as an intrinsic regulator of tumor cell survival. We show that STING regulates a transcriptional program that controls the generation of reactive oxygen species (ROS), and that STING loss alters ROS homeostasis to reduce DNA damage and to cause therapeutic resistance. In agreement with these data, analysis of tumors from head and neck squamous cell carcinoma patient specimens show that low STING expression is associated with worse outcomes. We also demonstrate that pharmacologic activation of STING enhances the effects of ionizing radiation in vivo, providing a rationale for therapeutic combinations of STING agonists and DNA-damaging agents. These results highlight a role for STING that is beyond its canonical function in cyclic dinucleotide and DNA damage sensing, and identify STING as a regulator of cellular ROS homeostasis and tumor cell susceptibility to reactive oxygen dependent, DNA damaging agents.

[1] Department of Therapeutic Radiology, Yale University School of Medicine, New Haven, CT, USA. [2] Department of Pathology, Yale University School of Medicine, New Haven, CT, USA. [3] F-Star Therapeutics, Cambridge, MA, USA. [4] Molecular, Cellular and Developmental Biology, Yale University, New Haven, CT, USA. [5] Department of Medicine, Yale School of Medicine, New Haven, CT, USA. [6] Department of Pharmacology, Yale University, New Haven, CT, USA. ✉email: Joseph.Contessa@yale.edu

DNA-damaging agents such as chemotherapy (e.g. cisplatin) and radiation therapy are fundamental treatment modalities for the majority of malignancies, however, development of therapeutic resistance remains a significant problem that leads to tumor recurrence and death[1–3]. These therapies induce DNA double-strand breaks (DSBs) and other DNA lesions[1–3], which has led to intense study of DNA repair proteins as potential targets for enhancing these treatments[4]. Conversely, targets that alter cellular environments to enhance or reduce DNA damage are less well understood. While an array of enzymes such as peroxidases, catalases, and super oxide dismutases are known to directly influence free radical homeostasis[5], few upstream regulators other than the Keap1/NRF2 antioxidant sensing pathway[5,6] have been identified in tumors, thus limiting strategies to manipulate this essential determinant of DNA damaging agent efficacy.

Genetic screens have been employed to functionally interrogate specific biological processes and for target identification in diverse biological systems[7–9]. Until recently genome-wide screens in mammalian cells have typically utilized RNA interference (RNAi) based methodologies. However, these approaches are limited, frequently with partial target suppression or off-target effects that lead to false-positive or negative results[10,11]. Specifically, it has been shown that off-target depletion of RAD51 is a common source of false positives from RNAi and as such RNAi is not well-suited to the study of DNA-damaging treatments such as radiation[12]. While there has been successful identification of previously unrecognized molecules involved in regulating DNA damage, these studies have typically been performed with RNAi or have utilized DNA-repair focused libraries, thus limiting the discovery of DNA damage effectors to known components of the DDR (e.g. canonical DNA repair proteins)[12–14]. CRISPR (clustered regularly interspaced short palindromic repeats)-Cas9 based genetic engineering has provided significant improvements for whole-genome screens due to more complete target suppression and fewer off target effects[7,8,15,16]. We therefore conducted an unbiased, genome wide CRISPR-Cas9 screen using radiation as a selective pressure for cell survival in order to identify novel targets and mechanisms for enhancing cancer therapies.

The endoplasmic reticulum-localized adaptor, STING (stimulator of interferon genes), is a critical regulator of the innate immune response through its ability to sense DNA damage via its recognition of cyclic dinucleotides and activate transcription of interferons and other cytokines[17,18]. While STING plays a role in the regulation of immune-mediated (extrinsic) responses to DNA damage via control of the CD8 + T-cell response, its role in controlling tumor cell intrinsic accumulation of DNA damage remains relatively uncharacterized[19–22]. Using genome-wide CRISPR-Cas9 screening we have made the unexpected discovery that STING loss alters redox homeostasis in tumor cells, identifying STING as an actionable target for reducing therapeutic resistance.

## Results

### Whole-genome CRISPR-Cas9 screen to identify novel regulators of DNA damage.
We performed a CRISPR-Cas9 screen using ionizing radiation as a selective pressure to identify novel genetic determinants of DNA damage. We generated Cas9-expressing FaDu head and neck squamous cell carcinoma (HNSCC) cells and verified that the Cas9 endonuclease does not alter cellular sensitivity to DNA damage (Supplementary Fig. 1a, b). It has been reported that multiple rounds of selection pressure at a dose that reduces survival to ~50% can enhance the sensitivity of genetic screens[23], therefore the radiation dose that killed 50% of cells by clonogenic survival analysis (2 Gy) was used for these experiments. FaDu-Cas9 expressing cells were screened with the human Genome-Scale CRISPR Knockout (hGeCKO v2) library containing 123,411 individual gRNAs (guide RNAs) targeting 19,050 genes[15]. The overall schema of the screen is shown in Fig. 1a. Briefly, irradiated cells were treated with four daily doses of 2 Gy and cells were collected 14 days after the final radiation dose to allow for cell death to occur. Genomic DNA was isolated and gRNAs were PCR amplified and sent for next-generation sequencing. More than 90% of gRNAs were detected in all samples (Supplementary Fig. 1c). Using Model-based Analysis of Genome-wide CRISPR-Cas9 knockout (MAGeCK) analysis[24], TMEM173 gRNA enrichment with an FDR-corrected P-value of 0.005 was found for irradiated vs. unirradiated samples (Fig. 1b and Supplementary Table 1). TMEM173 encodes the protein STING, a critical component of the innate immune machinery with a well-recognized role in regulating the immune-mediated and extrinsic response to radiation treatment in tissues[20,25]. The enrichment of gRNAs in the CRISPR-Cas9 screen suggests that STING also has an intrinsic role in regulating tumor cell sensitivity to DNA damage.

### STING controls tumor cell survival to DNA damaging agents.
STING expression modifies the immune-mediated anti-tumor effects of ionizing radiation in vivo[20,25]. A mechanism for STING's control of DNA damage is not known, though an indirect link through disruption of the cell cycle has been suggested[26]. To test the hypothesis that STING intrinsically regulates tumor cell sensitivity to DNA damage, we constructed three independent CRISPR-Cas9 STING knockouts (KO) in FaDu cells utilizing three individual gRNAs (Fig. 1c). Dose response clonogenic survival analysis with repeated doses of ionizing radiation (1–4 treatments of 2 Gy), similar to our screening regimen, showed that STING KO significantly enhances cell survival (Fig. 1e and Supplementary Fig. 1d). To confirm that these findings were due to STING KO, rather than an off-target effect, we performed genetic complementation experiments with STING over-expression in FaDu STING KO cells, and our results show that STING over-expression (Supplementary Fig. 1f) restored tumor cell sensitivity to ionizing radiation (Supplementary Fig. 1g). The role of STING in regulating cell survival after radiation exposure was also tested in the Detroit562 cell line using CRISPRi. Detroit562 cells with dCas9 expression were transduced with either a STING-targeting or non-targeting gRNA, and STING silencing was confirmed by immunoblotting (Fig. 1d). Clonogenic survival analysis was then performed with radiation treatment regimens described above. STING silencing (Fig. 1f and Supplementary Fig. 1e) also significantly enhanced cell survival after radiation treatment.

Tumor cell survival after cisplatin treatment, a DNA damaging chemotherapy used in HNSCC, was also tested. STING KO significantly enhanced resistance to cisplatin, particularly in the setting of multiple drug doses (~3–7-fold increase in resistance P < 0.04) as measured by clonogenic survival (Fig. 1g, h). In contrast cell survival after treatment with cetuximab, an EGFR antibody also used in HNSCC, showed that STING loss had no effect on cell death caused by blockade of kinase signaling (Fig. 1i–j). Together these results indicate that loss of STING confers resistance to DNA damaging therapies.

### STING regulates tumor radiosensitivity in vivo.
Stromal STING expression is required for maximal response to high-dose single fraction radiation treatment via interaction with CD8 + T cells[20]. However, the contribution of tumor STING expression to the in vivo radiation response has not been determined. Given our data suggesting that STING expression affects tumor cell survival after radiation exposure, we therefore used T-cell deficient

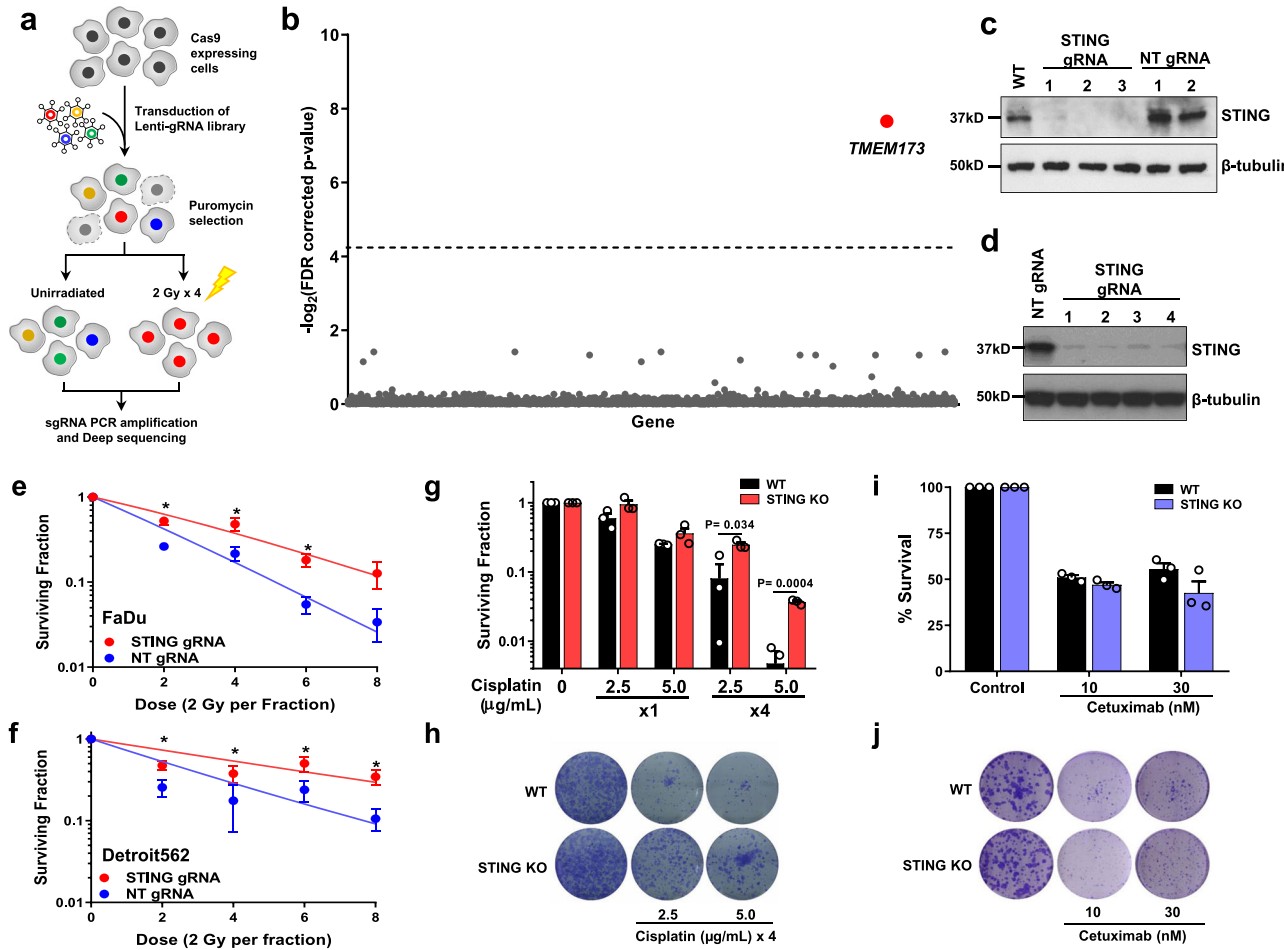

**Fig. 1 STING loss confers resistance to DNA-damaging agents. a** Schematic of CRISPR-Cas9 screen in FaDu cells aimed at identifying regulators of DNA damage. **b** Scatter plot showing genes corresponding to gRNAs that were significantly enriched in irradiated populations ($n = 3$ independent experiments) using MAGeCK analysis. **c** Immunoblot of FaDu isogenic STING knockout cells constructed with three independent STING gRNAs and two non-targeting (NT) gRNAs. Immunoblots are representative of two independent experiments. **d** Immunoblot of Detroit562 isogenic STING knockdown (dCas9) cells. Clonogenic survival analysis of FaDu STING KO (**e**) or Detroit562 STING silenced (**f**) cells treated with the indicated dose of ionizing radiation (2 Gy per fraction/day). Immunoblots are representative of two independent experiments (NT gRNA 1 and STING gRNA 4) or one independent experiment for STING gRNAs 1–3. In **e**, **f** error bars represent SEM from three independent experiments. In **e** $P$-values for 2, 4, and 6 Gy are 0.0002, 0.0002, and 0.04 as determined by unpaired, two-tailed $t$-tests without multiple comparison correction. In **f** $P$-values for 2, 4, 6, and 8 Gy are 0.03, 0.049, 0.01, and 0.02 as determined by unpaired, two-tailed $t$-tests without multiple comparison correction. Quantification (**g**) and representative images (**h**) of clonogenic survival analysis of FaDu WT and STING KO cells treated with Cisplatin as indicated. Error bars represent SEM from three independent experiments. analyzed by unpaired, two-tailed $t$-tests without multiple comparison correction. Quantification (**i**) and representative images (**j**) of clonogenic survival analysis of FaDu WT and STING KO cells treated with cetuximab. Error bars represent SEM from three independent experiments.

athymic nude mice to minimize contributions of the adaptive immune tumor response and tested the effects of STING loss on tumor growth after treatment with ionizing radiation in vivo. Mice-bearing STING WT (with a non-targeting gRNA) or STING KO tumors were randomized to receive fractionated radiation or no radiation as depicted in Fig. 2a. The growth curves for FaDu tumors corresponding to each treatment are shown in Fig. 2b. While the growth of WT and STING KO FaDu tumors were no different, irradiated STING KO tumors showed faster growth kinetics consistent with enhanced survival after radiation-induced DNA damage. Tumor growth delay, calculated as time to reach a volume of 600 mm³ (i.e. tumor tripling), in STING WT tumors was ~3-fold longer than STING KO tumors (Fig. 2c; 31.1 vs 10.43 days; $P = 0.019$). We also tested the effect of radiation on tumor growth in Detroit562 STING silenced xenografts (Fig. 2d). Consistent with the results from FaDu xenografts, loss of STING protein expression in Detroit562 tumors conferred significant resistance to radiation treatment with ~3-fold longer

growth delay compared to STING expressing tumors (Fig. 2e; 16.9 vs 5.1 days; $P < 0.0001$). Of note, one control tumor did not regrow and we assumed growth by Day 30 to provide a conservative estimate of the difference. In summary, these data indicate that in the absence of an intact immune system, loss of tumor cell STING expression confers resistance to ionizing radiation.

**STING regulates induction of DNA damage.** Radiation induces cell death primarily via induction of DNA DSBs and we sought to determine whether STING KO alters the amount of DNA damage using an analysis of γH2AX foci, shown to correspond to DNA DSBs[27]. We found that STING deficient cells have a significantly lower proportion of γH2AX foci positive cells 6 or 24 h after radiation exposure in both FaDu (57.4% vs 38.9%; $P = 0.004$ and 35.3% vs 21.3%; $P = 0.003$ respectively; Fig. 3a, b) and Detroit562 cells (85.8% vs 68.5%; $P = 0.005$ and 68.1% vs 34.5%; $P = 0.0004$

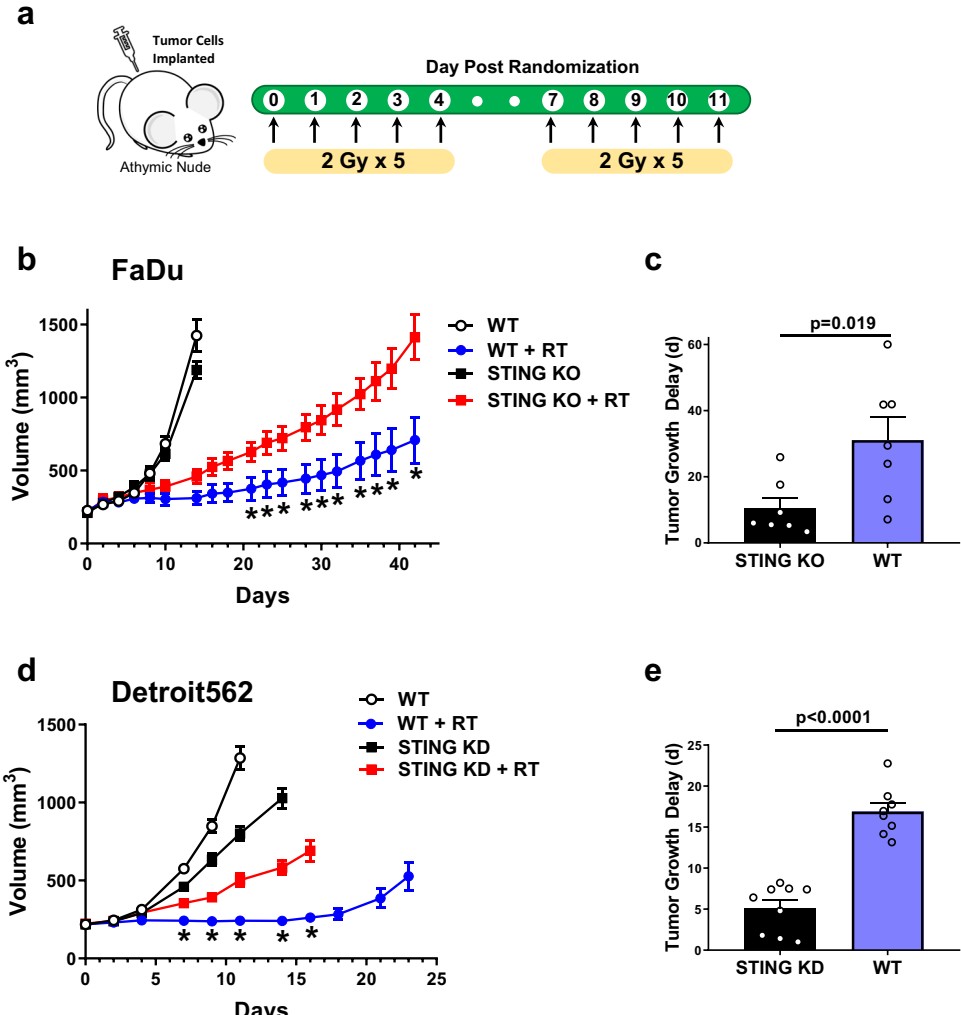

**Fig. 2 STING controls the in vivo response to radiation. a** Schematic of in vivo tumor growth delay experiments and fractionated treatment protocol with ionizing radiation. **b** Tumor growth curves for tumors generated from WT or STING KO FaDu cells implanted subcutaneously in athymic nude mice (error bars represent SEM for $n = 6$ for unirradiated groups and $n = 7$ for irradiated groups). *$P$-values for 21, 23, 25, 28, 39, 32, 35, 37, 39, and 42 days are 0.04, 0.02, 0.01, 0.005, 0.003, 0.0007, 0.0003, <0.0001, <0.0001, and <0.0001 based on two-way ANOVA with Fisher's LSD post-hoc analysis without multiple comparison correction. **c** Quantification of tumor growth delay from radiation treatment (time to reach 600 mm$^3$) from **b** (error bars represent SEM of $n = 7$ WT tumors and $n = 7$ STING KO tumors; $P$-value from unpaired, two-tailed Student's $t$ test). **d** Tumor growth delay curves from WT or STING silenced Detroit562 tumors (error bars represent SEM of $n = 9$ tumors for STING KD + RT group and $n = 8$ for all others). *$P$-values for 7, 9, 11, 14, and 16 days are 0.049, 0.002, <0.0001, <0.0001, and <0.0001 based on two-way ANOVA with Sidak post-hoc multiple comparison correction. **e** Quantification of tumor growth delay from radiation treatment (time to reach 600 mm$^3$) from **d** (error bars represent SEM of $n = 8$ WT tumors and $n = 9$ STING KD tumors; $P$-value from unpaired, two-tailed Student's $t$ test).

respectively; Fig. 3c, d). These results show that loss of STING expression is directly associated with higher levels of DNA damage. In agreement with these findings, analysis of DNA double strand breaks using the neutral comet assay demonstrated statistically significant reductions of tail moment at baseline and immediately after radiation exposure for STING KO cells (24.1 vs 17.6; $P = 0.012$ and 43.3 vs 33.4; $P = 0.024$), as well as a ~3-fold higher reduction at 24 h after radiation exposure (33.2 vs 12.2; $P < 0.0001$; Fig. 3e, f), demonstrating that STING KO causes a reduction of DNA double strand breaks.

Radiation-induced cell death as a consequence of DNA damage in non-hematopoietic cells typically occurs via mitotic catastrophe[28], which is evidenced by the presence of multiple distinct nuclear lobes within a cell[29–31]. Given the significant decrease in persistent DNA damage after radiation in STING KO cells, we examined cellular markers of mitotic catastrophe after radiation exposure. Indeed, STING loss significantly reduced the proportion of cells displaying multiple nuclear lobes (10.1% vs 19.5% at 72 h; $P = 0.002$; Fig. 3g, h). Although, STING has been reported to affect the distribution of cells in the S phase of the cell cycle[26], no cell cycle changes were observed in this CRISPR-Cas9 KO model system (Supplementary Fig. 2a) at baseline or after radiation. Together these data show that STING loss regulates cell survival by reducing the amount of DNA damage.

**STING regulates ROS homeostasis.** STING has a well characterized role in the innate immune response via its transcriptional control and activation of several downstream signaling pathways (e.g. TBK1/IRF-3 and ISG15)[17,32–34]. We therefore performed RNA-Sequencing (RNA-Seq) in our FaDu cells with and without STING KO to identify alterations in STING-dependent transcription that contribute to improved cell survival after exposure to ionizing radiation. DeSeq2 analysis showed that

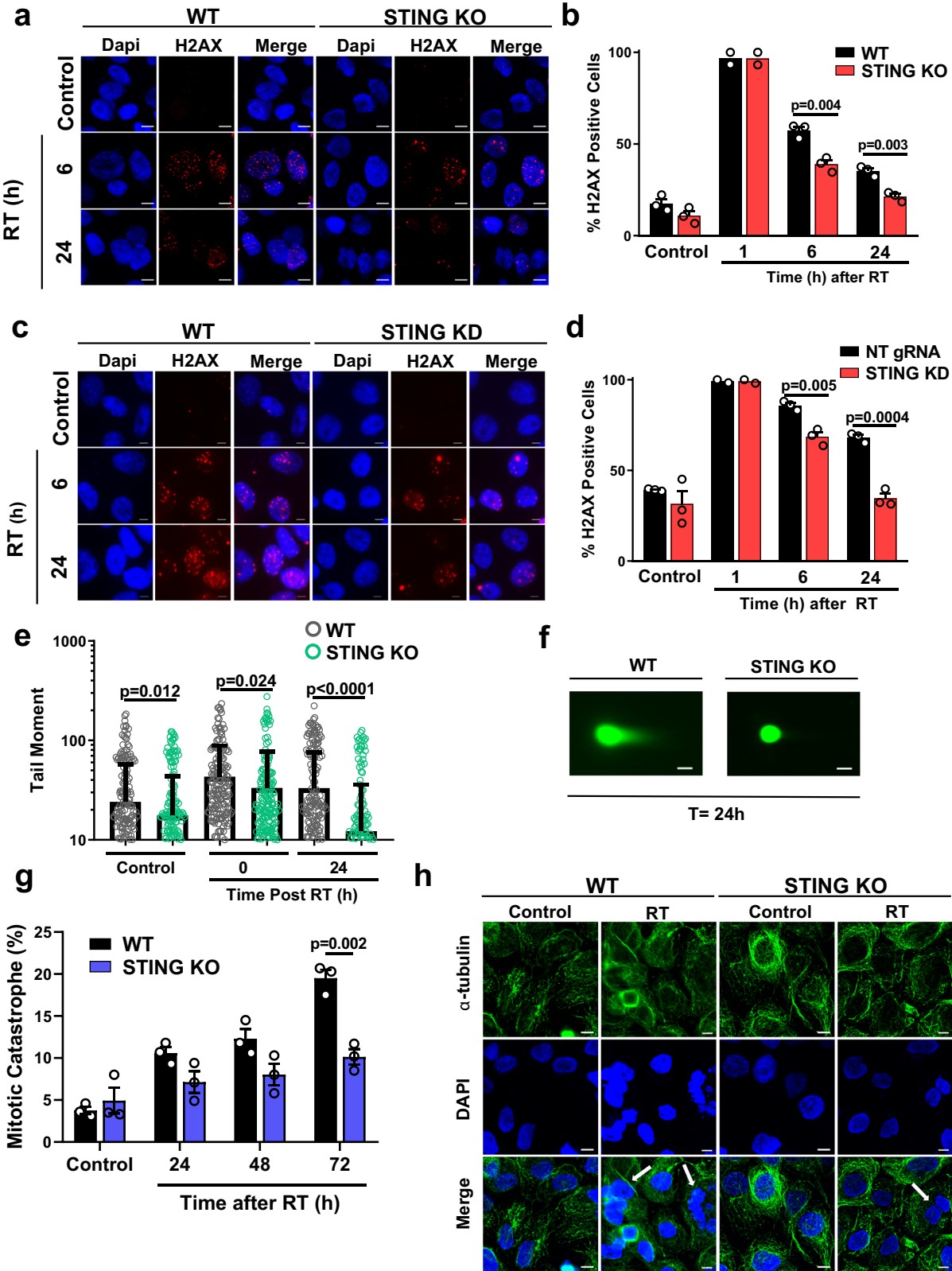

over 1000 genes were significantly altered in STING KO as compared to STING WT cells (Supplementary Data 1; FDR-corrected *P*-value < 0.05), with significant enrichment of genes involved in the interferon gamma pathway (Fig. 4a). Analysis of the top 150 downregulated genes is shown in Fig. 4b, and STING KO reduced both basal and radiation-induced expression of these transcripts (Supplementary Data 2). At the protein level STING

dependent signaling was enhanced by radiation with both induction of ISG15 levels and enhanced TBK-1 phosphorylation, effects that were eliminated by STING loss in both the Detroit527 and FaDu HNSCC cells (Fig. 4c, d). ISG15 is one of several genes (e.g. HERC5, KLF4, DUOX2)[5,35,36] identified by RNA-seq that is involved in regulation of reactive oxygen homeostasis[35], and Panther-based gene-ontology analysis of the 1075 genes

**Fig. 3 STING loss reduces radiation-induced DNA damage.** Representative images (**a**) and quantification (**b**) of radiation-induced γH2AX foci in WT and STING KO FaDu cells at the indicated times after RT (1 Gy × 4). Representative images (**c**) and quantification (**d**) of radiation-induced γH2AX foci in WT and STING KD Detroit562 cells at the indicated times after RT (1 Gy × 4). In **a** and **c** scale bares are 10 μm. In **b** and **d** error bars represent SEM from 2 (1 h post RT) or 3 (0 Gy, 6 and 24 h post RT) independent experiments, analyzed by unpaired, two-tailed *t*-tests without multiple comparison correction. Quantification (**e**) and representative images (**f**) of neutral comet assay foci performed in WT and STING KO FaDu cells at the indicated times after RT. Error bars in **e** represent SD for at least 195 cells from three independent experiments, scale bar is 30 μm, analyzed by unpaired, two-tailed *t*-tests without multiple comparison correction. Quantification (**g**) and representative images (**h**) of radiation-induced mitotic catastrophe in WT and STING KO FaDu cells at the indicated times after RT (1 Gy × 4). Arrows highlight examples of cells with multiple distinct nuclear lobes, a marker of mitotic catastrophe. Error bars represent SEM from three independent experiments, scale bar is 10 μm, analyzed by unpaired two-tailed *t*-test with Bonferroni–Sidak multiple comparison correction.

significantly affected by STING KO showed enrichment of genes involved in ROS-related pathways (e.g. "regulation of reactive oxygen species metabolic process"; $P = 0.006$). Given this downregulation of genes involved in ROS pathways by STING KO, we measured differences in radiation-induced ROS in our HNSCC cell line models. Using CM-H2DCDFA as a marker of cellular ROS, we found STING KO suppressed radiation-induced ROS generation by a factor of ~3 (5185 vs 14,500 at 24 h; $P < 0.0001$; Fig. 4 e, f). Glutathione peroxidase (GPX), an anti-oxidant molecule repressed by ISG15 and responsible for the reduction of $H_2O_2$ to water[37] was also shown to have increased activity by ~1.5-fold in STING KO cells compared to controls following radiation treatment, providing a direct mechanism for decreased ROS levels after STING KO (Fig. 4g, h). Collectively, these data indicate that STING is an upstream regulator of cellular ROS.

To directly examine the relationships between STING loss, ROS production, and DNA damage, we treated cells with either the ROS scavenger, *N*-acetyl cysteine (NAC), or with hydrogen peroxide ($H_2O_2$; Fig. 4i). Measurements of ROS after $H_2O_2$ treatment were significantly reduced in STING KO cells. This finding demonstrates a change in redox homeostasis and greater capacity for STING KO cells to metabolize ROS, which is consistent with our observation of elevated GPX activity following STING KO (Fig. 4g, h). Indeed, $H_2O_2$ treatment alone caused significantly more DNA damage, as measured by the number of γH2AX positive cells, in STING WT vs KO cells (Fig. 4j). In the setting of radiation treatment, NAC treatment reduced radiation-induced γH2AX levels of STING WT cells to that of STING KO cells. Conversely $H_2O_2$ treatment of STING KO cells increased radiation-induced γH2AX levels to those seen in the STING WT cells, demonstrating that ROS supplementation is sufficient to restore the extent of DNA damage. Together these results indicate that STING regulates transcriptional programs that suppress ROS metabolism, and thus primes the cellular environment for DNA damage initiated by $H_2O_2$ or ionizing radiation.

**STING expression levels are associated with HNSCC outcomes.** HNSCC patients are frequently treated with DNA damaging therapies (i.e. radiation and cisplatin), and because our data show that low STING expression is associated with therapeutic resistance, we sought to probe HNSCC patient outcomes dichotomized by STING levels. First we analyzed STING expression in the TCGA HNSCC cohort and found that low STING mRNA is associated with worse overall survival (OS; Fig. 5a; $P = 0.03$). To further investigate these findings we used a second HNSCC patient cohort with primary site biopsies of oropharyngeal SCCs ($n = 52$; Supplementary Table 2) to analyze STING protein expression. Because a prior study did not find an association of STING with cancer specific survival (CSS), we used an AQUA-based fluorescent analysis in order to provide continuous scoring of protein expression in tissue samples[38,39]. In agreement with the RNA expression, low STING protein measured across the entire specimen was associated with significantly worse progression-free survival (PFS; Fig. 5b; $P = 0.007$).

For this AQUA analysis tumor specimens were co-stained for STING, cytokeratin (tumor marker), and DAPI (nuclear marker; Fig. 5c), and it was evident that STING expression could clearly be localized to both tumor and stromal compartments (Fig. 5c). Given the understanding of STING's extrinsic effects via enhancement of anti-tumor immune responses and our discovery that STING regulates tumor cell survival after DNA-damaging treatment through ROS, we analyzed patient outcomes based on tumor or stroma STING protein levels. The data show that low STING in either the tumor ($P = 0.012$) or stromal ($P = 0.025$) compartment is associated with a significantly worse progression-free survival (PFS; Supplementary Fig. 3f, g). This finding suggests that both tumor intrinsic and extrinsic STING protein levels may be factors associated with tumor progression. We therefore stratified tumor patients into four groups based on high/low compartmental STING expression and found that the group with high STING in both the tumor and stroma had a significantly improved PFS (Fig. 5e). In addition, this compartmental analysis outperformed total STING levels (Fig. 5b) with a 5-year PFS of 78% vs 65%, suggesting that compartmental expression patterns should also be considered in future biomarker analyses. Overall, these patient data support the in vitro and in vivo findings that tumor cell STING levels are a significant determinant of cell survival following exposure to DNA damage. In addition, they suggest a strong rationale for investigating STING expression as a predictive biomarker.

**An intravenous STING agonist enhances the efficacy of radiation therapy.** Our data suggest that STING regulates the amount of DNA damage and thus tumor cell survival, a finding that is relevant for most patients with locally advanced HNSCC. We therefore asked whether STING activation with a next-generation, intravenous STING agonist (SB11285) would enhance efficacy of radiation in therapeutic models of HNSCC. Mice bearing FaDu or Detroit562 tumors were randomized into four treatment groups when tumors reached ~200 mm³: control (saline vehicle), radiation (2 Gy × 5), SB11285 (6 mg/kg), or the combination of radiation and SB11285 treatment was delivered. Radiation treatment or SB11285 alone had modest effects on FaDu tumor growth, however, the combination of SB11285 and radiation significantly inhibited tumor growth and prolonged the time to tumor doubling (median time to tumor doubling 7.7 vs 18.5 days for RT and RT + SB11285 respectively; $P < 0.0001$; Fig. 6a, b). From a toxicity perspective, the treatment was well tolerated with a transient, <10% weight loss associated with administration of SB11285 (Fig. 6c). A similar enhancement of radiation by SB11285 was observed in Detroit562 tumors (median time to tumor doubling 11.2 vs 17.1 days for RT and RT + SB11285 respectively; $P < 0.0001$; Fig. 6d, e). To examine the contribution of tumor STING expression to the efficacy of RT + SB11285 we used the same experimental design and treatment groups for the syngeneic MOC1 HNSCC tumor cell model implanted in C57BL/6J mice (Fig. 6f, g). Consistent with the literature, MOC1 cells have minimal STING expression (Fig. 6f)[40]. Radiation treatment or SB11285 alone had modest effects on tumor growth, and in contrast to our previous results, the combination of

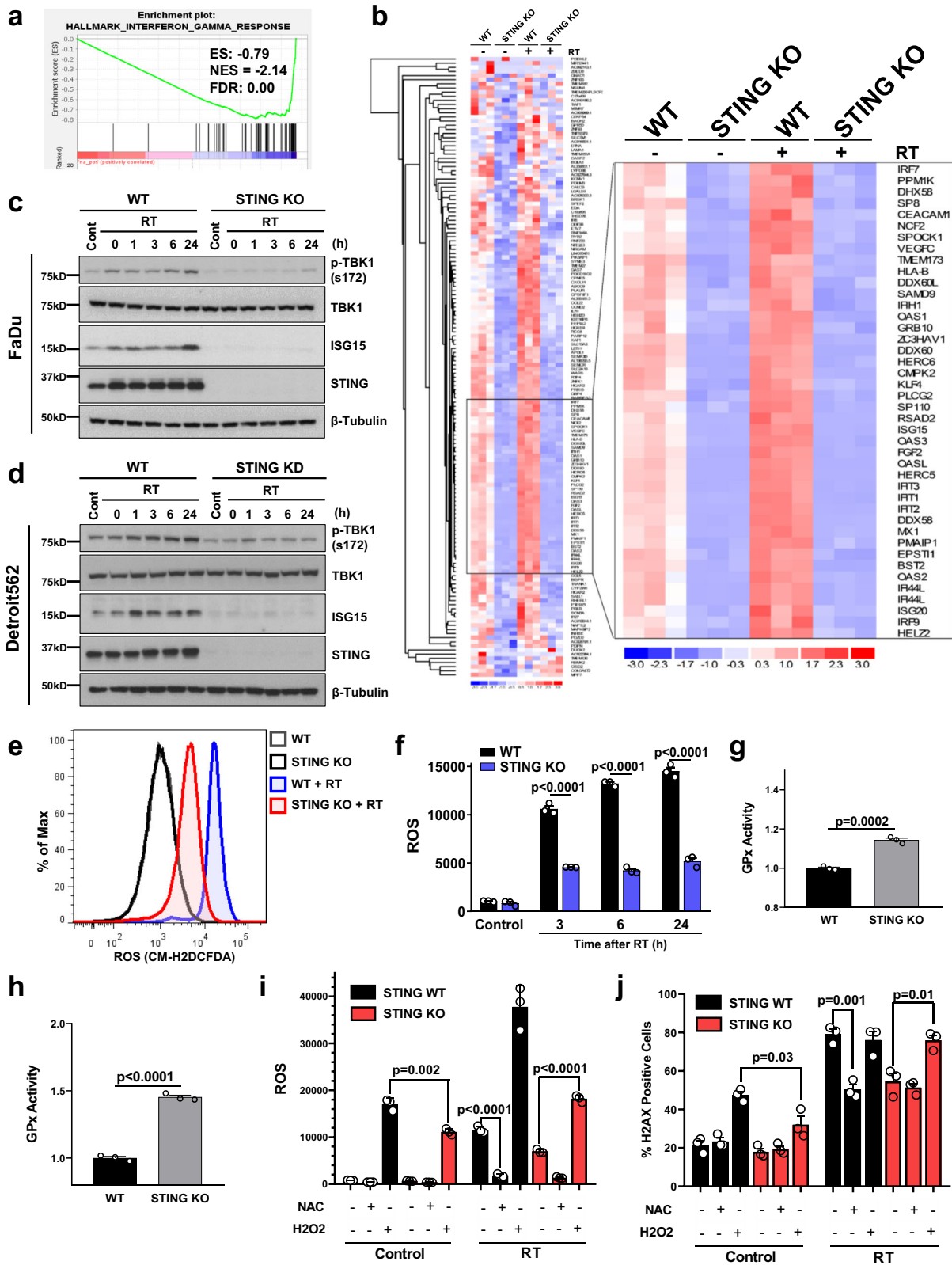

SB11285 with RT did not produce an additive delay in tumor growth relative to RT alone (Fig. 6g). Together these experiments demonstrate that systemic administration of a STING agonist in combination with radiation enhances local control in HNSCC and suggests STING expression in the tumor is required for maximal therapeutic effects. In summary we propose a model where STING loss represses the generation of ROS, reduces treatment-induced DNA damage, and results in inferior therapeutic responses (Fig. 6i).

## Discussion

To identify unrecognized genes that alter cellular survival in combination with DNA damaging agents and have the potential

**Fig. 4 STING loss alters the cellular transcriptome and ROS homeostasis. a** Gene set enrichment analysis (GSEA) plot of Hallmark IFNγ response in FaDu WT and STING KO cells. **b** Heatmap of the top 150 significantly downregulated genes in STING KO cells compared with WT. The inset identifies several critical genes in the ROS and ISG15 pathway with inhibited expression at baseline and prevention of radiation-induced expression. Immunoblots in FaDu (**c**) and Detroit562 cells (**d**) examining baseline and radiation-induced STING pathway activation for STING-regulated genes identified by RNA-Seq analysis. Immunoblots are representative of two independent experiments. Representative curves (**e**) and quantification (**f**) of flow cytometry-based analysis of ROS (measured by CM-H2DCFDA) in FaDu WT and STING KO cells. Error bars represent SEM from three independent experiments, analyzed by unpaired, two-tailed $t$-test without multiple comparison correction. Analysis of glutathione peroxidase (GPx) activity in FaDu WT and STING KO cells at baseline (**g**) and after radiation (**h**). In **g** and **h** error bars represent SEM from three independent experiments, analyzed by unpaired, two-tailed Student's $t$ test. **i** Flow cytometry-based analysis of ROS (measured by CM-H2DCFDA) in FaDu WT and STING KO cells treated with NAC (2 mM) or $H_2O_2$ (10 μM), or 6 h after RT (2 Gy × 4). Error bars represent SD from three independent experiments, analyzed by unpaired, two-tailed $t$-test. **j** Analysis of γH2AX foci in WT and STING KO FaDu cells treated with NAC (2 mM) or $H_2O_2$ (10 μM), or 6 h after RT (1 Gy × 4). Error bars represent SEM from three independent experiments, analyzed by unpaired, and two-tailed $t$-test without multiple comparison correction.

to inform new therapeutic strategies, we performed a whole-genome CRISPR-Cas9 screen with the selective pressure of ionizing radiation. Whereas previous efforts to perform similar screens have utilized RNAi-based approaches (either with whole-genome or targeted libraries), CRISPR-Cas9 has the advantage of more complete target suppression and potentially fewer off target effects[16]. Results from our screen showed an enrichment of *TMEM173*/STING KO, suggesting that STING loss enhances cell survival in the setting of DNA damage. In agreement, STING KO or silencing using CRISPR-Cas9 increased tumor cell resistance to both radiation and cisplatin treatment. RNA-Sequencing revealed that cellular transcription programs for ROS, the radicals that cause DNA damage, were reduced by STING KO, and associated with less DNA damage and enhanced tumor cell survival. To extend these findings to clinical scenarios we also examined the relationships between STING mRNA or protein expression and patient outcomes using the TCGA or a HNSCC tissue microarray (TMA), respectively. While both analyses are in agreement with our in vitro and in vivo studies and showed that low STING is associated with worse outcome, quantitative immunofluorescence of primary tumors specified that reduced STING expression in tumor cells is associated with tumor progression. This compartmental analysis thus provides further support for STING's tumor cell intrinsic role in mediating therapeutic response. In pursuit of translating these findings to preclinical models, we validated STING as an actionable target with SB11285, a small molecule STING agonist currently undergoing evaluation in clinical trials, and demonstrated enhancement of standard fractionated radiation therapy regimens. Together these results identify STING as a critical determinant of ROS homeostasis that alters tumor survival after radiation exposure, and also shows that activation of this pathway may be a principal strategy to enhance outcomes in HNSCC.

The canonical cyclic guanosine monophosphate (GMP)-adenosine monophosphate synthase (cGAS)-STING signaling pathway involves the recognition of cytosolic DNA by cGAS, the production of 2′-3′-cyclic GMP-AMP (cGAMP), and activation of STING leading to the transcription of type I IFNs and other cytokines[22,41–43]. For high dose, single fraction in vivo radiation treatment (and other DNA damaging agents), it has been shown that STING-dependent cytokines are important mediators of tumor cell killing[19,20,44,45], and that radiation-induced cell death is at least in part dependent on a CD8+ adaptive immune response[20]. The importance of the immune system is also supported by experiments in IFNAR1 (Interferon-α receptor 1) KO mice where anti-tumor effects of radiation are also reduced[19]. In addition, results from immunocompetent mouse models show that recognition of cGAMP by stromal STING is responsible for enhanced immune cell infiltration and immune-mediated cell death after radiation[21]. These results show that STING signaling plays an important role in the extrinsic (immune-mediated)

response to radiation. Our results diverge from these findings and highlight a mechanistically different role for STING, showing that it alters free radical homeostasis and induction of DNA damage per se. Although recent work has suggested that reduction of STING with RNAi leads to accumulation of cells in a cell cycle phase that is more resistant to DNA damage[26], our experiments using CRISPR-Cas9 to knock out STING function did not show changes in cellular proliferation or cell cycle distribution. Instead, our data show that STING loss imparts resistance to radiation through an increase in tumor ROS metabolism, and experiments using T-cell deficient immunocompromised mice support a role for STING's regulation of tumor cell intrinsic radiosensitivity. Our results are also in agreement with recent data suggesting that the cGAS-STING pathway is important in promoting mitotic cell death in response to taxanes[46] and in regulating sensitivity to TNFα upon BRCA2 inactivation[47]. Our study thus highlights a multi-faceted role for STING in regulating both the magnitude of DNA damage in addition to the promotion of an adaptive immune response.

Ionizing radiation induces DNA DSBs through two mechanisms; by ionization of electrons that directly interact with DNA or by indirect activation of electrons through generation of ROS species such as the hydroxyl radical. The indirect activation of electrons by ROS accounts for two-thirds of radiation-induced DNA DSBs and is therefore the critical mediator of cell death caused by ionizing radiation and select cytotoxic chemotherapies[5,48]. We show that STING modulates the expression of numerous interferon-related genes both at baseline and in response to radiation, with a subset of these STING-regulated genes involved in control of ROS. Consistent with these transcriptional changes we show that loss of STING leads to a reduction of ROS at baseline and after exposure to ionizing radiation or $H_2O_2$. The magnitude of this effect following radiation treatment (a reduction by 2–3-fold) is remarkable, predicts significant reduction in DNA damage, and is borne out by measurements of reduced γH2AX foci, comet tail moment, and markers of mitotic catastrophe in STING KO cells. Several studies have implicated components of the cGAS-STING signaling pathway in the control of DNA repair directly but with varying effect on cell survival[49–51]. In contrast our data show that rather than a direct effect on DNA repair, STING controls ROS homeostasis thus regulating the susceptibility for subsequent DNA damage. It has also recently been shown that GPX dependent ROS regulates STING directly, indicating a feedback loop that balances STING activity and expression with ROS homeostasis may be operative[52]. Our findings are highly relevant for cancer therapeutic strategies, and suggest that treatments reliant on ROS will be more effective in tumor cells with high levels of STING expression whereas therapies that do not rely upon ROS generation for their efficacy may be more relevant in the setting of STING loss.

In our preclinical models of HNSCC we show that loss of STING indeed causes resistance to ionizing radiation and cisplatin

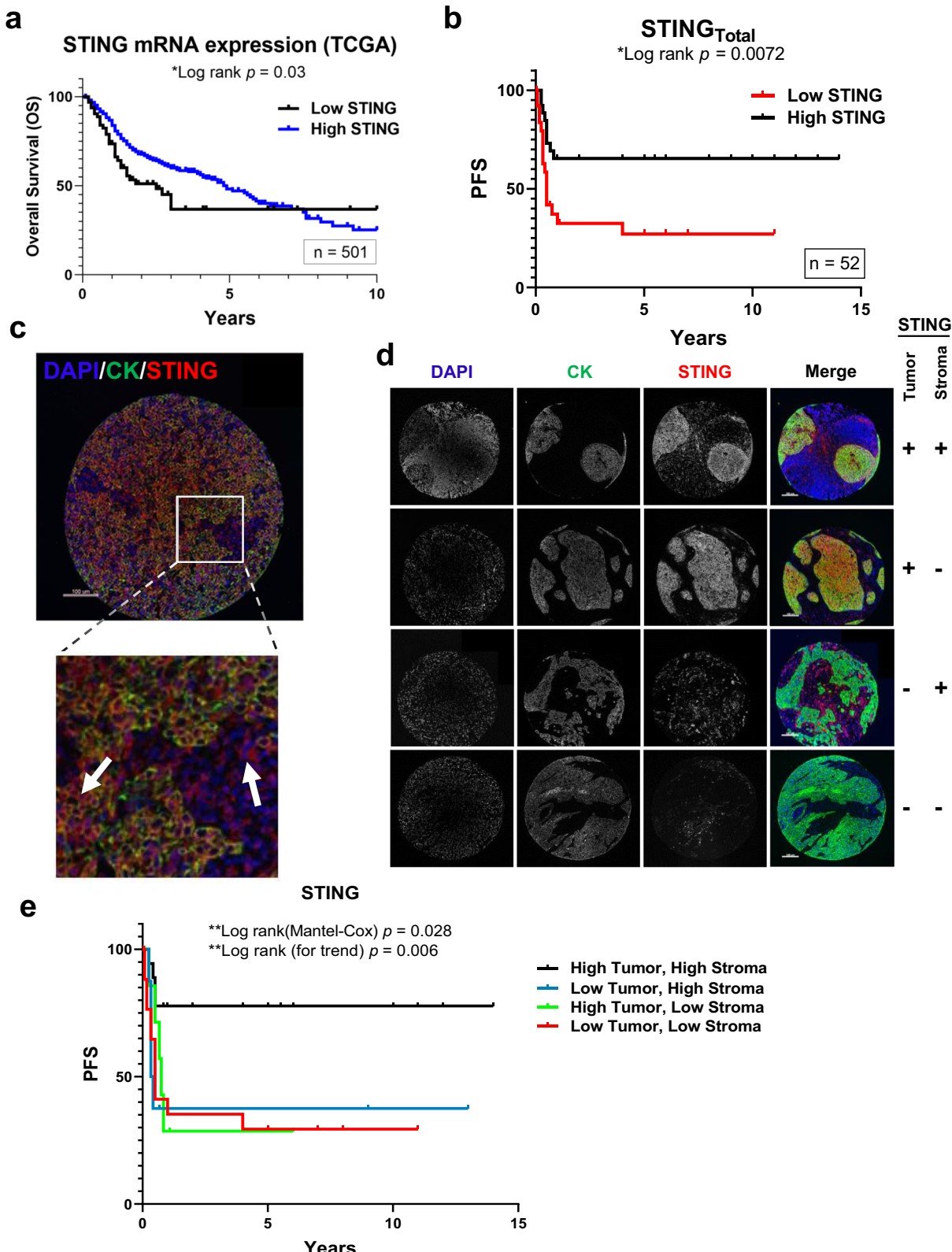

**Fig. 5 STING expression and outcomes in HNSCC. a** Kaplan–Meier curves of HNSCC TCGA cohort stratified by STING mRNA expression (low vs high). **b** Kaplan–Meier curves of patients stratified by total STING expression (low vs high). **c** Representative images of TMA spot from **b** stained for DAPI, cytokeratin (tumor mask), and STING. Arrows indicated STING staining in both compartments (tumor and stroma). **d** Representative images of TMA spots illustrating grouping of TMA specimens into four groups based upon compartmental STING expression. Each patient specimen was stained once due to lack of material, with images representative of their respective groups: **b** (from entire patient cohort) or **d** (groups defined in **e**). **e** Kaplan–Meier curves of patients stratified by four groups identified in **d**. All statistical analysis was performed by log-rank testing with *P*-values as indicated in graphs. Scale bars are 100 μm in **c** and **d**.

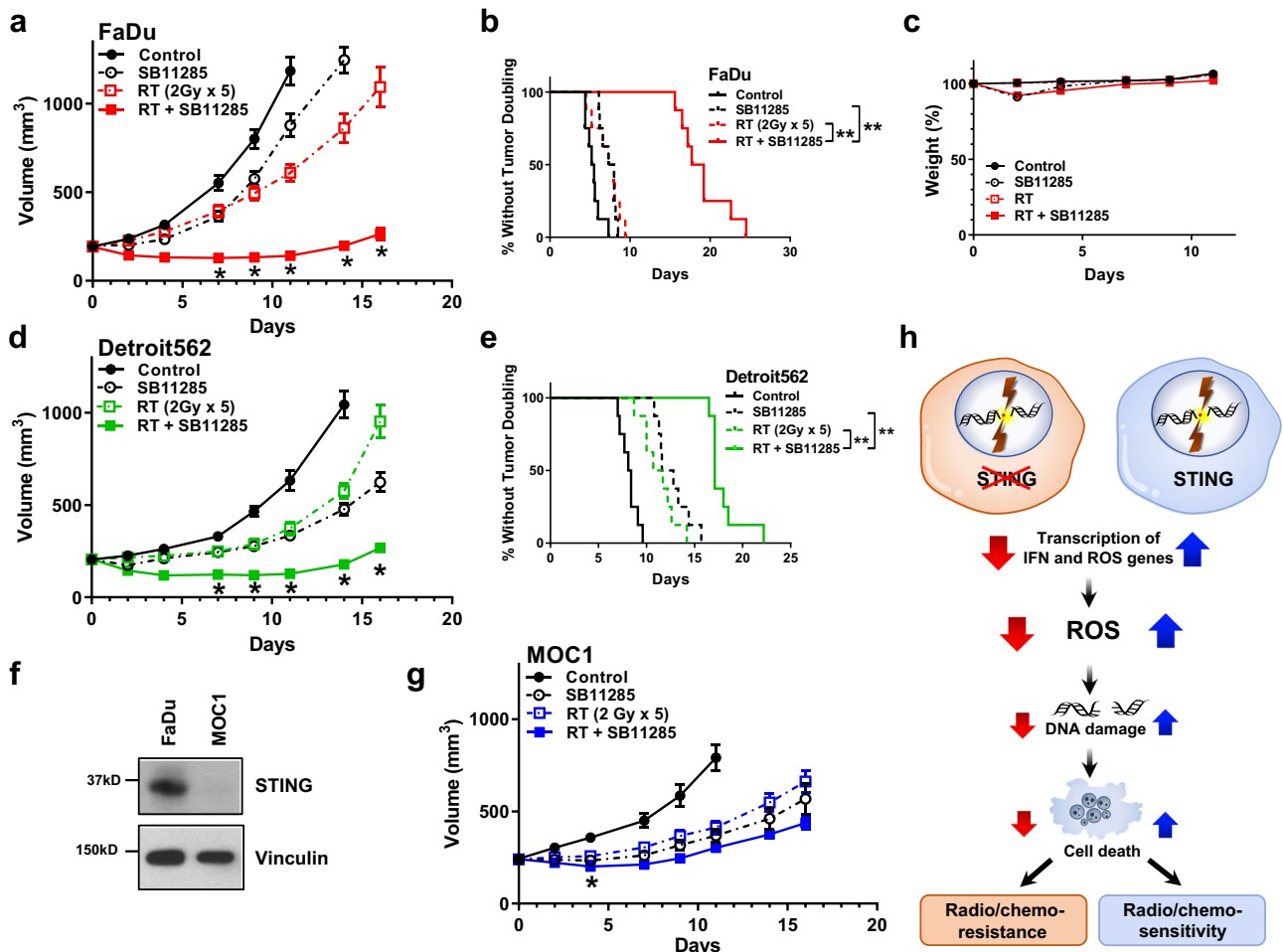

**Fig. 6 Novel STING agonist SB11285 enhances in vivo response to radiation therapy. a** FaDu tumor growth curves for each treatment group. Error bars represent the SEM of *n* = 8 mice per treatment group; *P-values for 7, 9, 11, 14, and 16 days are 0.0007, 0.0003, 0.0002, 0.0007, and 0.002 (RT + SBP11285 compared to RT) based on two-way ANOVA with Sidak post-hoc multiple comparison correction. **b** Kaplan–Meier curves of percent tumors without tumor doubling (400 mm³). Statistical analysis was performed by log-rank testing (*n* = 8 per treatment group), with **P < 0.0001. **c** Weight as stratified by treatment group (*n* = 8 mice per group). **d** Detroit 562 growth curves for each treatment group. Error bars represent the SEM of *n* = 8 mice per treatment group; *P-values for 7, 9, 11, 14, and 16 days are 0.001, 0.002, 0.002, 0.0002, and 0.002 (RT + SBP11285 compared to RT) based on two-way ANOVA with Sidak post-hoc multiple comparison correction. **e** Kaplan–Meier curves of percent tumors without tumor doubling (400 mm³). Statistical analysis was performed by log-rank testing (*n* = 8 per treatment group), with **P < 0.0001. **f** Western blots of STING expression in FaDu and MOC1 cells and are representative of two independent experiments. **g** MOC1 tumor growth curves for each treatment group. Error bars represent the SEM of *n* = 6 mice per treatment group; *P = 0.03 (RT + SBP11285 compared to RT) based on two-way ANOVA with Sidak post-hoc multiple comparison correction. **h** Proposed model of tumor intrinsic STING loss leading to treatment resistant phenotype.

treatment. To investigate the clinical relevance of these findings we examined outcomes for HNSCC patients first using STING mRNA levels in the TCGA dataset and second through an analysis of protein expression in an oropharyngeal SCC TMA using AQUA. Both cohorts show that lower integrated STING values from each sample type are associated with worse patient outcomes. Based on our new understanding of the intrinsic contributions of STING to tumor cell survival after treatment with DNA damaging agents, we undertook a compartmental analysis of STING protein expression that was afforded by tumor cell definition with cytokeratin and quantitative immunofluorescence. This analysis confirmed that low tumor cell STING levels alone (as well as low stromal levels alone) are associated with tumor progression, and support the importance of the tumor cell intrinsic function of STING signaling with respect to therapeutic resistance. We were unable to perform multivariate analysis in the TMA samples due to our patient cohort size (*n* = 52), which limited the statistical power to further characterize interactions of STING

with other clinicopathologic features (e.g. HPV status) and as such is the subject of future study. That patient outcomes were best differentiated by an analysis of compartmental STING expression and not by an integrated value across the sample suggests that this approach may be superior for investigating the predictive and prognostic potential of STING as a biomarker. A second observation from our preclinical models is that STING KO does not alter the anti-tumor effects of therapeutic agents such as cetuximab, which has a mechanism of action independent of ROS. Considering that a significant percentage of tumors have low STING, our data suggest the need to consider the efficacy of ROS-independent therapies such as cetuximab in tumors with low STING expression, as well as to identify strategies specifically aimed at improving outcomes in these patients.

Activation of the STING pathway to stimulate the host immune response and to enhance tumor control has recently been pursued in preclinical models[53–55]. To this end, STING pathway agonists are being tested either as monotherapies or in

combination with immune checkpoint inhibitors in clinical trials. However, testing of STING pathway agonists as monotherapies, such as intratumoral delivery of MK-145, have not demonstrated significant responses[56]. Our study shows that concurrent delivery of SB11285, a cyclic dinucleotide-based STING agonist that is intravenously available and has recently begun evaluation in clinical trials, improves the efficacy of radiation therapy in HNSCC tumors. Our data are in line with previous studies that have shown enhancement of radiation effects with intratumoral STING agonist injection (e.g. cGAMP or RR-CDG)[20,57], but our findings of successful i.v. administration suggest a path for rapid integration into standard HNSCC radiation treatment regimens. Although previous work has shown a role for STING for only high dose, single fraction radiation treatment, we show marked effects by manipulating STING in combination with the lower daily doses (i.e. 2 Gy) routinely used in clinical management. While others have shown that the anti-tumor effect of high dose radiation is partially dependent upon an extrinsic and adaptive immune response (specifically through the actions of CD8+ T-cells[20,58]), we now reveal a tumor cell intrinsic role for STING in regulating DNA damage. In addition, from a biomarker perspective, our experiments in syngeneic tumors with low STING expression show no significant enhancement of radiation efficacy by SB11285 and suggest that combinations of radiation therapy and a STING agonist would be most efficacious in tumors with intact STING expression. Moreover our preclinical data indicate that clinical evaluation of STING agonists in combination with radiation therapy deserves attention.

To our surprise, the only statistically significant result in our screen was STING. Clearly there are other regulators of the cellular radioresponse (e.g. *ATM, BRCA1/2, PRKDC,* and *RAD51*) that were not identified by our screening effort[59]. Screens that have identified components of the DNA repair machinery have traditionally used single exposures of drug or radiation[13]. In contrast our screen was constructed in a manner that serially reduced the surviving fraction by 50%, and was based upon previous genetic screening efforts suggesting this enhances their sensitivity[23]. Our data show that under this selective pressure, STING is a dominant factor that regulates DNA damage in HNSCC. STING may also play a more significant role in regulation of DNA damage in the context of multiple repeated genotoxic insults at doses that are clinically relevant, and this experimental design difference thus provides a reasonable explanation for why STING has not been identified in prior screening efforts.

In summary, we have performed an unbiased-whole-genome CRISPR-Cas9 screen with ionizing radiation to identify novel regulators of DNA damage. Our results show that STING (a protein with an established role in sensing cyclic dinucleotides) is also a regulator of reactive oxygen homeostasis, and that STING loss leads to enhanced ROS metabolism and therapeutic resistance to DNA damaging agents. These findings are supported by analyses of HNSCC patient outcomes that show low STING levels correlate with worse outcomes, suggesting that STING should be investigated as a biomarker in tumor types routinely treated with DNA damaging agents. Finally, we also demonstrate that STING activation in vivo enhances radiation therapy, an insight that warrants evaluation of STING agonists with DNA-damaging therapies in clinical trials.

## Methods

**Cell lines and treatment.** The human head and neck squamous cell carcinoma cells lines FaDu and Detroit562 were obtained from American Type Culture Collection (ATCC). Both cell lines have been validated by STR profiling by ATCC. HEK293T cells used for lentiviral production were a kind gift from Dr. Ryan Jensen (Yale University, New Haven CT). MOC1 cells were purchased from Kerafast. FaDu, Detroit562, and HEK293T were cultured in DMEM supplemented with 10% FBS (Gibco, Life Technologies). MOC1 cells were cultured in IMDM with Ham's

nutrient mixture (Gibco, Life Technologies), supplemented with 5% FBS (Gibco, Life Technologies), 1% Penicillin-Streptomycin (Gibco, Life Technologies), 5 mg/L insulin (Sigma-Aldrich), 40 µg/L hydrocortisone, and 5 µg/L EGF (R&D Systems). All cells were cultured in a humidified incubator with 5% $CO_2$, and they were kept in culture no more than 6 months after resuscitation from the original stocks. Mycoplasma cell culture contamination was routinely ruled out using MycoAlert Mycoplasma Detection Kit (Lonza). Cell cultures were irradiated using a Precision X-Ray 320 kV orthovoltage unit at a dose rate of 2.3 Gy/min with a 2-mm aluminum filter (Precision X-Ray Inc). Fractionated radiation was delivered, consistent with our CRISPR screen and as is done clinically. All in vivo experiments used 5 daily fractions of 2 Gy (as is done in standard clinical radiation experiments). Doses for in vitro experiments were chosen so as to not saturate experimental output with respect to cell death and is indicated in the respective figure legend. Quality assurance was performed monthly using a P.T.W. 0.3 cm$^3$ ionization chamber calibrated to NIST standards and quarterly dosimetry using thermoluminescent dosimeter-based or ferrous sulfate-based dosimeters. Cells were treated with Hydrogen Peroxide ($H_2O_2$) (Sigma-Aldrich) diluted in sterile water or freshly prepared n-acetyl cysteine (NAC; Sigma-Aldrich) dissolved in sterile water at the indicated concentrations just prior to irradiation.

**Clonogenic survival analysis.** For fractionated (FaDu and Detroit562) radiation experiments, cells were irradiated as indicated and twenty-four hours after RT cells were washed, trypsinized, and plated at clonal density in six-well plates to determine clonogenic survival. For cisplatin treatment, cisplatin (dissolved in PBS) was added to cells in serum free media for 1 h, cells were then washed twice, and complete growth media was added. In all, 24 h after final cisplatin dose, cells were washed, trypsinized, and plated at clonal density in six-well plates. For cetuximab treatment, cetuximab containing complete growth media was added to cells at clonal density and left for the duration of the experiment. In total, 10–14 days after seeding plates were stained with 0.25% crystal violet in 80% methanol. Colonies with >50 cells were counted. The surviving fraction of each sample was calculated as the ratio between the number of colonies counted divided by number of cells seeded and the plating efficiency, thus normalizing for plating efficiency differences with treatment (e.g. gene silencing/knockout). Representative images displayed in Fig. 1h–j have been taken to ensure the same number of cells seeded per well between the STING WT and STING KO conditions in that treatment group (e.g. same number of cells in 10 nM cetuximab-treated STING WT and STING KO wells). Clonogenic survival differences for each treatment were compared using survival curves generated from the linear quadratic equation as previously described[60].

**Immunoblot analysis.** Immunoblot analyses were performed as previously described[61]. Primary antibodies are listed in Supplementary Table 3 with their respective concentrations. Nitrocellulose-bound primary antibodies were detected with anti-rabbit IgG horseradish peroxidase-linked antibody or anti-mouse IgG-horseradish peroxidase-linked antibody (EMD-Millipore) and detected by Amersham ECL detection reagent (GE Healthcare).

**CRISPR-Cas9 screening.** The human GeCKOv2 CRISPR knockout pooled library was a gift from Feng Zhang (Addgene # 1000000049) and generated as described previously[15]. For each replicate of the pooled screen ~$1.6 \times 10^8$ Cas9-expressing cells were transduced at a MOI of ~0.3 with lentivirus produced by co-transfecting HEK293T cells with the packaging plasmids psPAX2 and pMD2.G (Addgene #12260 and 12259) and the aforementioned CRISPR-KO (GeCKOa and GeCKOb) library followed by selection with 2 µg/mL puromycin (Gibco) for 7 days. After puromycin selection pooled cells were irradiated (2 Gy daily for 4 days) with a Precision X-Ray 320 kV orthovoltage unit. In total, 14 days after the final radiation treatment cells were collected with unirradiated cells from the same experiment used as a control. Genomic DNA isolation was performed using QIAamp DNA Blood Columns (Qiagen, Hilden, Germany) and gRNA sequence were amplified using a two-step PCR reaction as described in Chen et al.[9] with maintenance of ~400x coverage of the GeCKO library. All PCR reactions were performed using Phusion Flash High Fidelity Master Mix (Thermo, F548L). Primers and barcode sequence are listed in Supplementary Table 4. Sequencing was performed with Illumina HiSeq single-end 75 bp reads. Reads were aligned to index sequences using the Bowtie aligner, and a maximum of one mismatch was allowed in the 20-bp gRNA sequence. The number of uniquely aligned reads for each library sequence was calculated after alignment for each of the three biologically independent replicates. Differential gRNA expression was analyzed in R using the Model-based Analysis of Genome-wide CRISPR/Cas9 Knockout (MAGeCK) method[24]. Briefly the MAGeCK algorithm normalizes read counts using median normalization followed by mean variance modeling to capture the relationship of mean and variance in the replicates. The statistical significance of each sgRNA is determined by using the learned mean-variance model). Robust rank aggregation (RRA) is utilized to determine essential or enriched genes and a FDR-corrected *p* value is determined. FDR-corrected *p*-value ≤ 0.05 was considered statistically significant.

**CRISPR-Cas9 and dCas9 cell line generation**. Cells were transduced with lentivirus containing Cas9 (lentiCas9-Blast; Addgene #52962) or dCas9 (lenti-dCas9-KRAB-blast; Addgene #89567). gRNAs were cloned into lentiGuide-Puro (Cas9 system; Addgene #52963) or pU6-sgRNA EF1Alpha-puro-T2A-BFP (dCas9 system; Addgene #60955) as previously described[15,62]. gRNA sequences are listed in Supplementary Table 4. Lentivirus was produced as above, and cells were transduced followed by puromycin selection. STING protein expression loss was confirmed by immunoblotting. STING gRNA 2 (FaDu) and 4 (Detroit562) and Non-targeting gRNA 2 (FaDu) were used for all downstream experiments Figs. 2–4 and Supplementary Figs. 2 and 3.

**Cell cycle analysis**. Cell-cycle distribution was determined by flow-cytometric analysis. Briefly, cells were treated as indicated, fixed with 70% ethanol, stained with FxCycle PI/RNAse staining solution (Invitrogen), and analyzed using a BD Bioscience LSR II flow cytometer. At least 10,000 cells per condition were analyzed with data shown representing the mean of three independent experiments.

**STING re-expression**. A STING cDNA in the pUNO3 vector was obtained from InvivoGen (San Diego, CA). The STING ORF was PCR amplified with a 5′ BamHI and 3′ NotI restriction site and cloned into pLV-EF1a-IRES-Neo (Addgene # 85139). Lentivirus was produced by co-transfecting HEK293T cells with the STING ORF containing vector the packaging plasmids psPAX2 and pMD2.G (Addgene #12260 and 12259). STING-KO cells were transduced and selected with 800 µg/mL G418.

**Immunofluorescence analysis of H2AX foci**. Cells were grown in chamber slides (Thermo Fisher), treated as indicated, fixed in 4% neutral buffered formaldehyde, permeabilized with 0.1% Triton X-100, and blocked with 1% bovine serum albumin in PBS-tween containing 5% goat serum. Slides were incubated with antibody to phospho-H2AX (1:500, Millipore) followed by incubation with goat-anti-mouse Alexa555 (1:750, Invitrogen) and mounted with Prolong gold antifade reagent with DAPI (Invitrogen). Cells were analyzed on a Leica SP5 confocal microscope with ×63 objective or EVOS M5000 Fluorescent Microscope with ×63 objective. Cells with 10 or more γH2AX foci were scored as positive[51,63–65]. Data presented are mean ± SEM of 2–3 biological replicates with >50 cells scored per experimental condition as indicated in the figure legend.

**Markers of mitotic catastrophe**. Cells were grown in chamber slides (Thermo Fisher), treated as indicated, fixed in 4% neutral buffered formaldehyde, permeabilized with 0.1% Triton X-100, and blocked with 1% bovine serum albumin in PBS-tween containing 5% goat serum. Slides were incubated with α-tubulin antibody (1:1000, Sigma-Aldrich) followed by incubation with goat-anti-mouse Alexa488 (1:1000, Invitrogen) and mounted with Prolong gold antifade reagent with DAPI (Invitrogen). Cells with nuclear fragmentation, defined as the presence of two or more distinct nuclear lobes within a single cell were defined to be undergoing mitotic catastrophe as previously reported[29–31,66]. Data presented are the mean ± SEM of three independent experiments with >50 cells scored per experimental condition.

**Neutral comet assay**. Neutral comet assay was performed according to the manufacturer's protocol (Trevigen). Briefly, cells were treated with RT (three daily fractions of 2 Gy followed by a single 8 Gy fraction on the final day), trypsinized, washed with PBS and suspended in LM Agarose (Trevigen). Electrophoresis in neutral conditions was conducted at 21 V for 1 h in the CometAssay Electrophoresis System (Trevigen). Data were collected with an EVOS M5000 Fluorescent Microscope. Data were analyzed using OpenComet software[67]. Data presented are mean ± SD for over 195 cells from three independent experiments.

**In vivo tumor growth delay**. In all, 4–6-week-old female athymic nude mice (Foxn1^nu, Envigo; FaDu and Detroit562 xenografts) or C57BL/6J (The Jackon Laboratory; MOC1 xenografts). Tumor xenografts were established by injection of $5 \times 10^6$ (FaDu and Detroit562) or $10 \times 10^6$ (MOC1) tumor cells subcutaneously into the right hind leg. When tumors reached ~200 mm³ animals were randomized into the specified groups. Radiation was delivered locally using a Siemens 250 kV orthovoltage unit at a dose rate of 6.42 Gy/min with a 2-mm aluminum filter (Siemens). Quality assurance was performed monthly using a P.T.W. 0.3 cm³ ionization chamber calibrated to NIST standards and quarterly dosimetry using thermoluminescent dosimeter-based or ferrous sulfate-based dosimeters. SB11285 was dissolved in normal saline and delivered via tail vein injection for FaDu and Detroit562 experiments or intraperitoneal injection for MOC1 experiments at the indicated dose. A single dose of SB11285 was delivered just prior to the first radiation treatment. Tumor size was measured three times per week and calculated according to the formula $(L \times W^2)/2$. Tumor growth delay was calculated as the time to reach a specified volume (e.g. 400 mm³ or 600 mm³ which represent tumor doubling or tripling) in the treated mice relative to untreated mice as has been previously described[65,68] and as such accounts for differences in baseline tumor growth. Data are expressed as a mean ± SEM tumor volume. Group sizes are specified in the respective figure legends. All experimental procedures were approved in accordance with IACUC and Yale University institutional guidelines for animal care and ethics and guidelines for the welfare and use of animals in cancer research.

**Detection of intracellular ROS**. Cells were plated in 10 cm dishes and irradiated as indicated. Adherent cells were incubated with 5 µM CM-H2DCFDA (5-(and-6)-chloromethyl-2′7′-dichlorodihydrofluorescein diacetate acetyl ester; Invitrogen) for 30 min at 37 ℃, then the cells were washed once with PBS. Stained cells were collected by trypsinization and resuspended in PBS. ROS generation was assessed by flow cytometry (excitation, 488 nm; emission, 515–545 nm) with $2 \times 10^4$ cells for each condition.

**Determination of glutathione peroxidase activity**. The activity of glutathione peroxidase (GPx) was evaluated by colorimetric assay according to manufacturer's protocol from Abcam (GPx Activity Kit; ab102530). In brief, $2 \times 10^6$ cells were collected and depleted of all GSSG by incubating the sample with glutathione reductase (GR) and reduced glutathione (GSH) for 15 min. GPx activity was determined by adding cumeme hydroperoxide and incubating for 0 and 5 min. The absorbance was determined at OD340.

**Patient cohort and tissue microarrays**. We analyzed retrospectively collected, formalin-fixed, and paraffin-embedded (FFPE) tumor specimens which were in TMA format. Specimens were collected and used with specific consent or waiver of consent under the approval from the Yale Human Investigation Committee protocol #9505008219. The HNSCC cohort (YTMA329) contained 186 oropharynx tumors resected between 2001 and 2012 from both primary and metastatic lymph node sites. For our analysis, we included only the patients in the cohort who had primary site specimens and removed from analysis samples resected from metastatic lymph node sites, leaving us with specimens from a total of 52 patients. Detailed clinicopathologic information of the patients analyzed is presented in Supplementary Table 2. We built a custom 'index' TMA (YTMA419) for reagent titration, assay validation, and reproducibility assessments. This index TMA contained FFPE cores of normal kidney tissue and both normal and cancerous breast tissue which were used for negative STING controls and cores of normal lymph node tissue and HNSCC samples for positive STING controls. The TMA also contained FFPE prepared parental FaDu cells and FaDu STING KO cells. Cell-line TMA construction has been published in detail elsewhere[39]. Please see supplementary materials for further details.

*Materials*. SB11285 was obtained through a research agreement with Spring Bank Pharmaceuticals (now F-Star Therapeutics).

**TCGA analysis**. RNA-sequencing data and the clinical metadata with a total of 546 samples in read counts (HTSeq-Counts) of head and neck cancer were obtained from the TCGA data portal (https://portal.gdc.cancer.gov/). The clinical information of the 546 samples was filtered to exclude the samples from solid tissue normal and only the samples from primary tumor and metastatic samples ($n = 501$) were analyzed. X-Tile cut-point finder software was used to determine thresholds to define low and high STING expression in TCGA head and neck cancer cohort. Overall survival (OS) curve was constructed using the Kaplan–Meier analysis with a follow-up of 10 years and statistical significance was determined using the log-rank test.

**Statistical analysis**. Results are expressed as mean ± SEM unless otherwise indicated. GraphPad Prism 7.0 software (GraphPad software, Inc., La Jolla, CA) was used for statistical analysis as described within Results. $P$-value ≤ 0.05 was considered statistically significant. All tests are two-tailed unless otherwise indicated.

**Reporting summary**. Further information on research design is available in the Nature Research Reporting Summary linked to this article.

## Data availability
CRISPR screening data (GSE147084) and RNA-sequencing data (GSE147085) to the Gene Expression Omnibus (GEO). Further data associated with this study are available by reasonable request to the corresponding author. Source data are provided with this paper.

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

## Acknowledgements
Research reported in this publication was supported by the Radiological Society of North American Resident Research Grant RR1752 (T.J.H.), the National Cancer Institute of the National Institutes of Health under Award Number K12CA215110 (T.J.H.). The content is solely the responsibility of the authors and does not necessarily represent the official views of the National Institutes of Health. This work is also supported by the Yale SPORE in Head and Neck CancerP50 DE030707 (J.N.C., D.L.R, B.A.B.), the Yale Cancer Center Pilot and Discovery Grants (J.N.C.), the Yale SPORE in Lung Cancer, and the Yale Cancer Center (D.L.R.). This study was also funded in part by a research agreement with Spring Bank Pharmaceuticals (now F-Star Therapeutics; J.N.C.).

## Author contributions
T.J.H., B.A.B, D.L.R., and J.N.C. conceived the study and contributed to scientific hypothesis. T.J.H., M.B., T.M., C.P., W.C., D.L.R., and J.N.C. contributed to experimental design and methodology. T.J.H., M.B., T.M., C.P., T.N.A, W.C., and T.S.S. performed experimental work. R.I, K.L, and S.C determined in vivo STING agonist dosing. T.J.H., D.L.R, and J.N.C. interpreted the data. T.J.H. and J.N.C. wrote the manuscript. All authors reviewed the manuscript.

## Competing interests
J.N.C. has a research agreement with Spring Bank Pharmaceuticals. D.L.R. has served as a consultant, advisor, or served on a Scientific Advisory Board for Amgen, Astra Zeneca, Agendia, Biocept, BMS, Cell Signaling Technology, Cepheid, Daiichi Sankyo, GSK, Merck, NanoString, Perkin Elmer, PAIGE, Sanofi, and Ultivue. He has received research funding from Astra Zeneca, Cepheid, Nanostring, Navigate/Novartis, NextCure, Lilly, Ultivue, and Perkin Elmer. B.A.B. received an honorarium from Aduro. K.L., R.I., and S.C. were employed by Spring Bank Pharmaceuticals/F-STAR. All remaining authors declare no competing interests.
