## [Peer Review File · Nature Communications]

Reviewers' comments:

Reviewer #1 (Remarks to the Author):

The manuscript by Hayman et al reports on a CRISPR-Cas9 screen for gene deletions that trigger resistance to ionizing radiation (IR). The screen is performed in head and neck squamous cell carcinoma (HNSCC) cells and the only hit is STING, a key factor of the DNA-mediated innate immune response. The results are solid, as IR resistance upon STING inactivation is recapitulated in other cell lines and with multiple gRNAs. Importantly, tumour xenografts carrying STING deletion are resistant to IR, relative to STING wt counterparts, which points out to a clear clinical potential for STING levels to predict tumour responses to radiotherapy.

The authors investigate next the mechanism by which STING inactivation can suppress IR sensitivity. Their results indicate that STING acts as suppressor of DNA double-stranded break (DSBs) repair. Thus, in STING KO cells DSB repair is enhanced relative to STING wt cells. STING has been known for a while to facilitate DNA-mediated innate immune responses (Ishikawa&, Nature 2009) and to be activated by IR-induced DNA damage (Harding et al., Nature 2017). The results presented in this manuscript imply that STING potentiates IR-induced DNA damage accumulation and is a key mediator of IR toxicity. Work by Liu et al. (Nature 2018) demonstrated that cGAS, a factor acting upstream of STING in the response to cytosolic DNA damage, inhibits the homologous recombination pathway of DSB repair. It is therefore not surprising that STING, which is activated by cGAS, also suppresses DNA repair. Hayman et al. however do not identify the repair pathway(s) inhibited by STING. This could be easily done using the several assays available in the field.

In addition, the authors suggest that in response to IR, STING protects mitotic cells from death by mitotic catastrophe. These data (Fig. 3H) are rather unconvincing, as the quantification criteria are not clear ('multiple nuclear lobes'). For an example of cell death by mitotic catastrophe quantification, the authors should refer to Aarts et al. (Cancer Discovery 2012) or Zierhut et al (Cell 2019).

The authors then present the results of RNA-seq analyses performed before and after IR, in STING wt and KO cells. Top deregulated genes are, as expected, components of the STING- and interferon-dependent pathways. Uniquely, genes involved in regulation of reactive oxygen species (ROS) are specifically downregulated in STING KO cells. Thus, the authors claim that STING is required for the production of IR-induced ROS, but their results fail to convince. The authors should strengthen them for example by comparing STING activity in ROS control with ribonucleotide reductase, recently shown to regulate this process (Somyajit et al., Science 2017).

One key question that the authors fail to address is how ROS downregulation correlates with suppression of DNA repair in STING KO cells.

Finally, the authors show that among HNSCC patients, low STING protein levels predict poor outcome in terms of progression-free survival. Conversely, stimulating STING with the agonist SB11285 enhances the anti-tumour effect of IR. These results are consistent with the model proposed by the authors according to which STING is a key mediator of IR response and of its tumour suppressive effect.

In summary, this is an interesting paper reporting the first CRISPR screen for genes that prevent IR resistance, performed in the specific context of head and neck cancer cells. IR treatment is critical for head and neck cancer patients. STING is the only hit isolated from this screen. The clinical data presented in the paper support STING as a mediator of IR toxicity in cells and tumours. However, the mechanism is only partially addressed and should be strengthened before publication.

Reviewer #2 (Remarks to the Author):

In their manuscript entitled: 'STING Enhances Cell Death Through Regulation of DNA Damage and Reactive Oxygen Species' Hayman and co-workers describe a role for STING in the response to radiation. Using a CRISPR-based screen, STING was identified as a determinant of viability in response to irradiation. These findings were confirmed, and appeared to also be valid in the context of cisplatin treatment. Also, levels of gamma-H2AX remained high in Sting k.o. cells after irradiation, and these cells also showed higher levels of mitotic catastrophe. Moreover, a higher level of glutathione-peroxidase activity was demonstrated in Sting k.o. cells, which was interpreted as a key role for Sting in ROS regulation. In a separate set of experiment, sting was immunohistochemically analyzed in a TMA of head-and-neck cancers, and staining intensity was related to clinical data of patients. Finally, a sting agonist was used in combination with irradiation and showed significant potentiation of RT effects.

STING signaling has been connected to DNA damage in recent years, including in response to radiation-induced DNA damage. In recent literature, multiple reports have connected cGAS/STING signaling to the response to DNA damage, including DNA damage induced by irradiation. For instance, Harding et al reported activation of cGAS/STING signaling in response to irradiation (Nature, 2017; PMID: 28759889), Parkes et al showed that inactivation of DNA repair components or induction of S-phase DNA damage induced STING activation (Parkes et al, 2016; PMID: 27707838), and a genome-wide screen identified that inactivation of inflammatory signaling genes resulted in resistance to DNA damage induced by BRCA2 inactivation (Heijink et al, 2019; PMID: 30626869). The authors should do a better job in citing relevant literature, as none of these papers have been cited in their manuscript.

Overall, the described screen is interesting (although I weirdly only identified 1 hit). The follow-up experiments in my opinion do not show any mechanistic insight, whereas the text claims strong conclusions. The patient analysis and in vivo work using the Sting agonist show impressive differences, but these analyses are mechanistically not connected the Sting and a role in ROS regulation. To my opinion, the lack of mechanistic insight, combined with the poor design and description of experiments, makes that this paper is not suitable for publication in Nature Communications.

Comments;

- Mechanistically, it remains totally unclear what the role of STING is in mediating DNA damage tolerance. Sting is referenced to as a 'master regulator of ROS', but I don't see evidence for this statement, and no mechanistic insight is provided how STING does this. The only link is elevated activity of glutathione peroxidase and a loss of ISG15, but this could be indirect.
- If STING-mediated buffering of ROS is the suggested mechanism: treatment of ROS scavengers (such as NAC) should also turn cells resistant to IR. There is some evidence that lowering ROS levels also lowers IR-induced DNA damage, but this was not reported to block IR-induced cell death.
- The screen data is only described in very limited detail. Also, it remains unclear how the data was normalized to reach only FRD values as shown in the list. The screening data is not deposited as it seems.
- The analysis of clinical material: only a univariate analysis is performed. The analysis should include relevant clinicopathological features (age, stage, treatment, etc) to that a multivariate analysis can be performed. Importantly for head-and-neck cancers, HPV status should be included. Also, OS should be analysis in addition to PFS. This is an important etiological factor for these cancers, and has been linked to STING function (Lau et al, Science, 2015).
- Throughout the paper the treatment regimens change. Number of irradiation cycles, amounts of dose are different. Unclear how this affects the data.
- In the in vivo studies, the tumors are not immunohistochemically investigated for ROS, of DNA damage, which would link the observed effects in vivo to the suggested mechanisms.
- The clonogenic survival assays: unclear how the quantification is derived from the images. Compare 10 vs 30 nM cetuximab in images with bar graphs.
- It seems that the STING KD cells have a significant growth delay in vivo. this could have a major impact on IR sensitivity.

- The in vivo analysis of growth delay until 600 mm² is reached is very arbitrary.
- For the FaDu cell line all three clones are analyzed by WB, for the Detroit cell lines, only one clone is shown. Also, it is unclear which of the clones is used for which experiment.
- Mitotic catastrophe is only scored on DNA staining of fixed cells. It is unclear if these cells have undergone mitosis. This should either be done by live-cell microscopy, or include relevant markers (combined phospho-HistoneH3 with cleaved caspase3 for instance). Binucleated cells are typically not considered as mitotic catastrophe.
- The Sting re-expression experiments; include the parental cell line, to see if the re-expression is near endogenous or over-expressed.
- The reporting is not clear at multiple cases. The H2AX foci are shown as H2AX-positive cells versus negative cells. This is not acceptable, and should be reported as numbers of foci per cell, with individual cells as datapoints.
- RNAseq is not described in the methods. Data is not deposited, unclear with the number of replicates is. Analysis is very limited. What is the effect of IR in these cells at the RNA levels? Also, it appears very arbitrary how ISG15 is selected as a DEG.
- Page 3: CRISPR screens are common nowadays. The advantage over RNAi does not need to be introduced extensively.
- Line 76: STING is not a DNA sensor itself, but a downstream target of DNA sensors.
- Line 396: replicated -> replication

Reviewer #3 (Remarks to the Author):

Review of Hayman T et al. Nature communications
 STING Enhances Cell Death Through Regulation of DNA Damage and Reactive Oxygen Species
 NCOMMS-20-10341-T

Summary

The manuscript describes the discovery that STING is a regulator of radiosensitivity in cancer cells as a result of a whole genome CRISPR screen. Surprisingly, this was the only gene discovered using this tool, but STING lies at a very interesting intersection of DNA sensing and immune regulation. This is a novel discovery and has the potential to be impactful.

However, while the paper identifies STING as a regulated gene, it does not adequately prove mechanism. Instead it transitions to the use of STING ligands as a cancer therapeutic. This has been demonstrated in a few different manuscripts, so is of lesser value. However, the major issue in this transition is that exogenous addition of STING does not fit with the initially described mechanism of STING as part of the DNA damage response in the cancer cell. The paper would be more valuable if it were able to prove how STING impacts radiosensitivity.

The manuscript is well-written and the experiments seem appropriate for their conclusions. There are no significant statistical issues. Figures are clear and well presented.

Major issues

The studies stop short of providing mechanism. While loss of STING decreases radiosensitivity, and results in a loss of IFN pathway activation and ROS production, we do not know whether these are mechanistic. Since only STING was identified by the genetic screen, and not TBK1, IFN receptors, or ROS-related genes, it is difficult to interpret relevance of the fact that these pathways are activated. These data would suggest that these pathways are not critical to the effects of STING loss. Additional experiments identifying whether the IFN pathway or the ROS-generating genes are necessary or sufficient for the observed effects would be important to make this case.

The experiments using systemic STING ligands are difficult to interpret in view of the intrinsic DNA damage response model that is explored in the early part of the paper. While STING ligands can activate adaptive immunity in cancer models, the literature available thus far suggests that they

work via STING expressed in stromal cells – in particular macrophages and dendritic cells, but occasionally endothelial cells. This is distinct from the effects of STING activation in the cancer cells by radiation-driven production of endogenous cGas-generated cyclic dinucleotides. The authors make the case that STING expression in cancer cells is important to the response to radiation therapy, but they do not test STING treatment where the cancer cells or the stroma lack STING (either using STING^{-/-} mice or their STING^{-/-} cancer cells). It is plausible that the STING ligands are working through mechanisms that are independent of cancer cell expression of STING, as has been shown in other preclinical studies. Without requiring too many additional experiments or new mouse strains, it would be important to know whether RT+IV STING therapy remains effective if the cancer cells no longer express STING.

The clinical data of patient outcome based on STING expression is very interesting, but it is notable that both HPV+ and HPV- patients are part of the study. It would be important to know whether these cohorts are differently represented in the STINGhi or STINGlo tumor/stromal groups, since others have shown that HPV+ cancers tend to express more STING in cancer cells (PMID: 29135982), and HPV+ patients generally have more favorable outcomes.

Reviewer #1 (Remarks to the Author):

The manuscript by Hayman et al reports on a CRISPR-Cas9 screen for gene deletions that trigger resistance to ionizing radiation (IR). The screen is performed in head and neck squamous cell carcinoma (HNSCC) cells and the only hit is STING, a key factor of the DNA-mediated innate immune response. The results are solid, as IR resistance upon STING inactivation is recapitulated in other cell lines and with multiple gRNAs. Importantly, tumour xenografts carrying STING deletion are resistant to IR, relative to STING wt counterparts, which points out to a clear clinical potential for STING levels to predict tumour responses to radiotherapy.

The authors investigate next the mechanism by which STING inactivation can suppress IR sensitivity. Their results indicate that STING acts as suppressor of DNA double-stranded break (DSBs) repair.

We thank you for this review. Your critique makes it entirely clear that the manuscript did not effectively communicate the main point of our work- that STING regulates reactive oxygen species levels (ROS). Although DNA double strand break repair is a logical mechanism to study in the context DNA damaging agents, the novelty of our study is that we show an unexpected mechanism for regulation of DNA damage by STING. In retrospect, upon reviewing the prior version of the manuscript, we see how the presentation and language we used did not adequately tell our scientific story. First, we included a paragraph about DNA damage and repair in the introduction which suggested that is what we intended to study. Second we used vague terminology such as “the cellular response to DNA damage” which can be described more simply as survival but sounds like “the DNA damage response” ie DNA repair. In truth, there were instances where our word choice also introduced unintended claims that STING affected DNA damage repair. In the revised manuscript we have significantly reorganized the text and figures to focus on our discovery of a novel mechanism for STING’s regulation of ROS and cell survival, removed the introductory paragraph on DNA repair, and significantly reduced the use of ‘cellular response’ so as not to mislead the reader. In this new version we believe we clarified the intent of the study and the interpretations of the data to support this newly discovered role for STING.

Thus, in STING KO cells DSB repair is enhanced relative to STING wt cells. STING has been known for a while to facilitate DNA-mediated innate immune responses (Ishikawa&, Nature 2009) and to be activated by IR-induced DNA damage (Harding et al., Nature 2017). The results presented in this manuscript imply that STING potentiates IR-induced DNA damage accumulation and is a key mediator of IR toxicity. Work by Liu et al. (Nature 2018) demonstrated that cGAS, a factor acting upstream of STING in the response to cytosolic DNA damage, inhibits the homologous recombination pathway of DSB repair. It is therefore not surprising that STING, which is activated by cGAS, also suppresses DNA repair. Hayman et al. however do not identify the repair pathway(s) inhibited by STING. This could be easily done using the several assays available in the field.

We agree with the reviewer that this is a fertile area for further investigation. Since initial submission of the manuscript Dr. Hayman has accepted an Asst. Professor position and will take this project to his newly established lab where he seeks to investigate the potential for regulation of DNA repair by STING.

In addition, the authors suggest that in response to IR, STING protects mitotic cells from death by mitotic catastrophe. These data (Fig. 3H) are rather unconvincing, as the quantification criteria are not clear (‘multiple nuclear lobes’). For an example of cell death by mitotic catastrophe quantification, the authors should refer to Aarts et al. (Cancer Discovery 2012) or Zierhut et al (Cell 2019).

Thank you for this critique which was also raised by Reviewer 2. Review of the literature has demonstrated to us that there are indeed varying definitions for mitotic catastrophe. The definition we used, founded upon quantification of multi-nucleation, is accepted by many groups (and was recently used for a paper published in Nature Communications [29]). Our goal in pursuing this series of experiments was to show that the DNA damage we observed with gH2AX and COMET assays is also consistent with changes seen at the cellular level. It does not appear that the reviewers are challenging the data about multinucleation, rather they question whether this is definitively mitotic catastrophe. In either case, the data strongly

supports our observations that STING loss leads to less DNA damage and reduced appearance of multi-nucleate cells. To respond to the reviewers' critiques we have made the following changes 1) we state that multi-nucleation is a marker of mitotic catastrophe 2) we remove mention of mitotic catastrophe from the abstract, 3) we remove mention of mitotic catastrophe from the discussion, and 4) from the summation figure (Fig. 6h). We do keep this term as a label in Figure 3g for simplicity and believe we have provided enough information for the reader to accurately understand and interpret the data.

The authors then present the results of RNA-seq analyses performed before and after IR, in STING wt and KO cells. Top deregulated genes are, as expected, components of the STING- and interferon-dependent pathways. Uniquely, genes involved in regulation of reactive oxygen species (ROS) are specifically downregulated in STING KO cells. Thus, the authors claim that STING is required for the production of IR-induced ROS, but their results fail to convince. The authors should strengthen them for example by comparing STING activity in ROS control with ribonucleotide reductase, recently shown to regulate this process (Somyajit et al., Science 2017).

We agree with the reviewer that if we were focused on DNA repair, as was interpreted from the initial manuscript, this would be a potential route for investigating a mechanistic interaction. However, our focus is on upstream regulation of the amount of DNA damage that is produced when STING is present or absent, and we have instead undertaken additional mechanistic studies suggested by Reviewer 2 in order to show that ROS levels can be manipulated to recapitulate or rescue the cellular phenotypes produced by manipulation of STING at the genetic level.

One key question that the authors fail to address is how ROS downregulation correlates with suppression of DNA repair in STING KO cells.

This critique is in line with the other comments regarding DNA repair, and we hope that our revised and restructured manuscript makes it clear that we are not linking ROS to DNA repair.

Finally, the authors show that among HNSCC patients, low STING protein levels predict poor outcome in terms of progression-free survival. Conversely, stimulating STING with the agonist SB11285 enhances the anti-tumour effect of IR. These results are consistent with the model proposed by the authors according to which STING is a key mediator of IR response and of its tumour suppressive effect.

In summary, this is an interesting paper reporting the first CRISPR screen for genes that prevent IR resistance, performed in the specific context of head and neck cancer cells. IR treatment is critical for head and neck cancer patients. STING is the only hit isolated from this screen. The clinical data presented in the paper support STING as a mediator of IR toxicity in cells and tumours. However, the mechanism is only partially addressed and should be strengthened before publication.

We thank the reviewer for these comments which showed us that we were not successful in clearly explaining the novelty and impact of our work. Our resubmission contains significant revisions in the text, re-organization of the figures, and new mechanistic as well as animal data to strengthen our findings and to provide a strong rationale for both CRISPR based screening strategies to identify targets, and the translation of STING agonists to therapeutic settings involving DNA damaging agents.

Reviewer #2 (Remarks to the Author):

In their manuscript entitled: 'STING Enhances Cell Death Through Regulation of DNA Damage and Reactive Oxygen Species' Hayman and co-workers describe a role for STING in the response to radiation. Using a CRISPR-based screen, STING was identified as a determinant of viability in response to irradiation. These findings were confirmed, and appeared to also be valid in the context of cisplatin treatment. Also, levels of gamma-H2AX remained high in Sting k.o. cells after irradiation, and these cells also showed higher levels of mitotic catastrophe. Moreover, a higher level of glutathione-peroxidase activity was demonstrated in Sting k.o. cells, which was interpreted as a key role for

Sting in ROS regulation. In a separate set of experiment, sting was immunohistochemically analyzed in a TMA of head-and-neck cancers, and staining intensity was related to clinical data of patients. Finally, a sting agonist was used in combination with irradiation and showed significant potentiation of RT effects.

STING signaling has been connected to DNA damage in recent years, including in response to radiation-induced DNA damage. In recent literature, multiple reports have connected cGAS/STING signaling to the response to DNA damage, including DNA damage induced by irradiation. For instance, Harding et al reported activation of cGAS/STING signaling in response to irradiation (Nature, 2017; PMID: 28759889), Parkes et al showed that inactivation of DNA repair components or induction of S-phase DNA damage induced STING activation (Parkes et al, 2016; PMID: 27707838), and a genome-wide screen identified that inactivation of inflammatory signaling genes resulted in resistance to DNA damage induced by BRCA2 inactivation (Heijink et al, 2019; PMID: 30626869). The authors should do a better job in citing relevant literature, as none of these papers have been cited in their manuscript.

We thank the reviewer for highlighting additional literature that are relevant to STING and the DNA damage response and have added the following references to the manuscript:

1. Harding et al 2017, Nature, PMID 28759889 (Ref 22, Page 4, Par 2)
2. Parkes et al 2016, JNCI, PMID 27707838 (Ref 45, Page 15, Par 2)
3. Heijink et al, 2019, Nat Comms, PMID 30626869 (Ref 47, Page 16, Par 1)
4. Carozza et al 2020 Nature Cancer (Ref 21, Page 15, Par 2)
5. Moore et al 2016 Cancer Immunol, PMID 27821498 (Ref 40, Page 13, Par 1)
6. Mackenzie et al 2017 Nature, PMID 28738408 (Ref 42, Page 15, Par 2)
7. Kwon et al 2020 Cancer Disc, PMID 31852718 (Ref 43, Page 15, Par 2)
8. Jiang et al 2019 EMBO J, PMID 31544964 (Ref 49, Page 16, Par 2)
9. Chen et al 2020 Sci Adv, PMID 33055160 (Ref 50, Page 16, Par 2)
10. Liu et al 2018, Nature, PMID 30356214 (Ref 51, Page 16, Par 2)
11. Jia et al 2020, Nat Immunol, PMID 32541831 (Ref 52, Page 17, Par 1)
12. Liu et al 2019, Nat Comms, PMID 31704921 (Ref 55, Page 18, Par 2)

Overall, the described screen is interesting (although I weirdly only identified 1 hit). The follow-up experiments in my opinion do not show any mechanistic insight, whereas the text claims strong conclusions. The patient analysis and in vivo work using the Sting agonist show impressive differences, but these analyses are mechanistically not connected the Sting and a role in ROS regulation. To my opinion, the lack of mechanistic insight, combined with the poor design and description of experiments, makes that this paper is not suitable for publication in Nature Communications. Comments;

- Mechanistically, it remains totally unclear what the role of STING is in mediating DNA damage tolerance. Sting is referenced to as a 'master regulator of ROS', but I don't see evidence for this statement, and no mechanistic insight is provided how STING does this. The only link is elevated activity of glutathione peroxidase and a loss of ISG15, but this could be indirect.

Thank you for this comment. Similar to comments from reviewer 1, we now realize that the language used in the first version of the manuscript was imprecise. We are not suggesting that STING mediates tolerance—rather that it changes the cellular state itself with respect to ROS. We therefore describe this more clearly by stating that STING alters ROS homeostasis, which remains a novel and impactful finding. Based on the reviewer's comment we have also removed the term 'master' regulator, however our RNAseq data clearly demonstrates STING knockout changes the transcriptional profile of genes involved in ROS and therefore we do refer to it as a 'regulator' of this process. To provide evidence that transcriptional regulation does indeed have mechanistic consequences for ROS we measure changes in ISG15, a known negative regulator of GPX4, to show that ISG15 translation (and enhancement by radiation) is nearly eliminated by STING knockout, leads to enhancement of GPX activity, and reduces ROS. Additional mechanistic considerations are addressed below.

- If STING-mediated buffering of ROS is the suggested mechanism: treatment of ROS scavengers (such as NAC) should also turn cells resistant to IR. There is some evidence that lowering ROS levels also lowers IR-induced DNA damage, but this was not reported to block IR-induced cell death.

We thank the reviewer for this suggestion to mechanistically test our model regarding STING's role in regulation of ROS. We therefore performed a series of new experiments using addition of NAC (a ROS scavenger) or H₂O₂ (ROS) in the setting of STING knockout. We found these results to be extremely informative. First, we show that measurements of ROS after the addition of H₂O₂ to WT or STING KO cells demonstrates a significantly reduced amount of ROS in STING KO (Fig. 4i). This shows that in the absence of DNA damaging agents, STING KO cells have an increased ability to buffer exogenous ROS. This finding is entirely consistent with data that shows enhanced GPX activity (Figs. 4g,h). Second, we demonstrate that the addition of H₂O₂ alone is sufficient to increase DNA damage as measured by γH2AX positive cells, but that H₂O₂ causes less DNA damage in the setting of STING KO (Fig. 4j), and is consistent with increased ROS buffering (Fig. 4i) and increased GPX activity (Figs. 4g,h). In the setting of radiation exposure, we use NAC to reduce radiation-induced ROS to levels below those seen in STING KO (Fig. 4i). This manipulation also significantly reduces radiation induced DNA damage (Fig. 4j). However, free radical scavenging with NAC is unable to further reduce radiation-induced DNA damage in cells with STING KO (Fig. 4j) which have lower levels of radiation-induced ROS (Fig. 4i). We also examined the addition of H₂O₂ in combination with radiation exposure and found that there were no significant enhancements of DNA damage in WT cells (Fig. 4j). However, the addition of H₂O₂ significantly enhanced DNA damage in STING KO cells, effectively 'rescuing' the induction of DNA damage by radiation. Again, we thank the reviewer for suggesting this line of experimentation because it provides significant insights into the effects of STING KO on ROS buffering and homeostasis, demonstrates that two separate approaches for enhancing ROS (H₂O₂ treatment or radiation exposure) both induce more DNA damage in WT vs STING KO cells, and that ROS scavenging or addition of H₂O₂ can duplicate or rescue the effects of STING KO, respectively. We believe this series of experiments provides new mechanistic data that strongly supports our findings regarding the role of STING in regulating ROS homeostasis.

- The screen data is only described in very limited detail. Also, it remains unclear how the data was normalized to reach only FRD values as shown in the list. The screening data is not deposited as it seems.

We thank the reviewer for pointing out that that screening methodology should be made more clear. To remedy this problem, we have submitted both the CRISPR Screening data (GSE147084) and RNA-sequencing data (GSE147085) to the Gene Expression Omnibus (GEO) with these accession numbers added on [Page 30, paragraph 2]. We have also added additional language to the methods section to briefly describe the MAGECK methodology [Page 24, paragraph 1]. Additionally, should further details regarding the analysis be required they are described in full in Reference 24.

- The analysis of clinical material: only a univariate analysis is performed. The analysis should include relevant clinicopathological features (age, stage, treatment, etc) to that a multivariate analysis can be performed. Importantly for head-and-neck cancers, HPV status should be included. Also, OS should be analysis in addition to PFS. This is an important etiological factor for these cancers, and has been linked to STING function (Lau et al, Science, 2015).

We thank the reviewer for this critique which raises similar concerns to that of Reviewer 3. Here we admit that we should not have used the term 'prognostic' as that claim would require further analysis. This description derailed the logic and goal of analyzing and presenting this patient data. The goal of this figure is to show that our insights into the biology of STING- that low tumor expression reduces DNA damage and improves survival after treatment with DNA damaging agents- correlates well with patient outcomes. In this figure we advance this concept further by demonstrating that both tumor and stromal STING expression levels can be used to identify patient subsets that have better (or worse) outcomes.

With respect to biomarkers in HNSCC, validation has been a major challenge to the field. Although we now accept p16/HPV as prognostic, many early studies analyzing large numbers of patients were confounded by clinical prognostic factors such as age or tumor stage. In fact, validation of p16/HPV as a prognostic factor reported by Ang et. al (NEJM 2010) required analysis of a 720 patient randomized trial! The analysis of a clinical trial and establishment of STING as an independent prognostic factor is beyond the scope of this manuscript. We are, however, in collaboration with other groups to analyze their institution's HNSCC TMA and have also initiated parallel projects to perform secondary analysis of completed clinical trials. These are independent studies being led by David Rimm (a co-author and pathologist) and have required significant expansion of contributors and lead authors for these future works.

To respond to the reviewers' critiques, we now provide additional data to support the association of STING expression with patient outcome. The impetus for performing the TMA analysis of STING protein expression was based on our observation that low STING RNA expression levels are associated with worse survival in the TCGA dataset. We did not include this data in the first version of the manuscript because the protein data is more directly relevant to the issue of tumor cell STING expression, but now see that we should have included this secondary dataset as an independent patient cohort. The TCGA data is now added as Figure 5A (and below). Due to the size of this cohort (n=501 patients) we were able to analyze outcomes with respect to overall survival and the data shows that patients with low STING levels had a significantly worse overall survival. The low number of samples from our TMA (n=52) limited further analysis with respect to competing prognostic factors or the longer term endpoint of overall survival. When we revisited the TCGA dataset, we were surprised to learn even in this very large data set, the numbers of patients with known HPV status (n=103) were insufficient to perform an analysis of STING expression in HPV+ or HPV- patients (below). Thus, further analysis of larger patient cohorts is warranted and ongoing.

In summary, to address the reviewers' comments we have removed the term 'prognostic' and instead state that analysis of STING in larger cohorts is warranted, with discussion of limitations [page 17]. To provide confidence that this association is real, we now also show data from an independent patient cohort that demonstrates worse overall survival in patients with low STING expression.

In the revised manuscript we now further emphasize the importance of this clinical data with respect to our novel findings on STING's role in regulating ROS homeostasis. Figures 1-4 show that low STING expression alters an ROS transcriptional program, reduces ROS, reduces DNA damage, and enhances tumor cell survival. This data predicts that tumor cell STING levels are clinically important for sensitivity or resistance to DNA damaging agents. Figure 5a and 5b show that in HNSCC patients (often treated with DNA damaging agents), low STING at both the RNA and protein levels is associated with worse patient outcomes. Moreover, the clinical data from TMAs can be analyzed at a more granular level and supports

our findings that tumor cell STING levels (and not just immune cell levels) are important, as low STING in either tumor or stromal compartments is associated with worse PFS (Fig 5e) and outperforms integrated analysis of STING protein across the tumor sample (Fig. 5b).

- Throughout the paper the treatment regimens change. Number of irradiation cycles, amounts of dose are different. Unclear how this affects the data.

Thank you for bringing this to our attention. We have updated the manuscript with our rationale for selection of dose and number of fractions [Page 21, Paragraph 1]. Briefly, experimental fractionated radiation regimens are now detailed in the manuscript and are consistent with the fractionated radiation utilized in the CRISPR screen and are similar to doses delivered clinically. *In Vivo* experiments consistently used 2 Gy x 5 (one week of daily 2 Gy), similar to clinically delivered daily doses. Radiation doses (both *in vitro* and *in vivo*) were chosen so as to not saturate experimental outcomes with respect to cell death.

- In the in vivo studies, the tumors are not immunohistochemically investigated for ROS, of DNA damage, which would link the observed effects in vivo to the suggested mechanisms.

We agree with the reviewer that a direct measurement of ROS *in vivo* would provide an *in vivo* extension of our *in vitro* mechanistic work. However, the measurement of ROS directly in tumor samples remains a challenge for the ROS field and to our knowledge is not readily feasible. Nonetheless, we believe our expanded *in vivo* data with new experiments in a syngeneic model (Fig. 6g) provide additional data to strengthen our argument that STING controls cell death after radiation.

- The clonogenic survival assays: unclear how the quantification is derived from the images. Compare 10 vs 30 nM cetuximab in images with bar graphs.

We thank the reviewer for pointing this out and apologize that this was unclear. We have added additional verbiage to the clonogenic survival methods section to describe scoring and normalization [Page 22 Paragraph 2]. Representative images have been taken to ensure the same number of cells seeded per well between the treated and STING WT and STING KO conditions. As is standard with clonogenic survival assays a higher number of cells must be seeded initially to account for loss in cell viability with differing treatments (and this is taken into account with calculation of plating efficiency and surviving fraction as described). Thus, images with drug treatment (and increasing drug dose) were taken from wells with a higher number of cells seeded to account for drug induced toxicity (relative to untreated wells) and as such direct comparison between different drug doses are not useful visually. Rather these images are meant to depict differences between STING WT or KO cells with different treatments.

- It seems that the STING KD cells have a significant growth delay in vivo. this could have a major impact on IR sensitivity.

We thank the reviewer for the opportunity to clarify a few points regarding tumor growth delay calculations and measurements of radiation sensitivity. While STING loss appears to have a small baseline effect on tumor growth delay relative to WT tumors this difference is statistically significant only in the last time point in the FaDu xenograft experiment and in the last two timepoints of the Detroit562 comparison. Furthermore differences in baseline tumor growth are accounted for in the tumor growth delay measurements as described in the manuscript. Specifically, tumor growth delay is calculated as the time to reach a specific volume in the treated tumors relative to the untreated tumors. As such underlying differences in tumor growth at baseline are accounted for and will not confound radiation response measurements. To clarify this measurement we have added additional verbiage to the methods section [Page 27 and Paragraph 1].

- The in vivo analysis of growth delay until 600 mm2 is reached is very arbitrary.

We thank the reviewer for pointing out that our lack of explanation made this endpoint seem arbitrary. Radiation-induced tumor growth delay across treatment groups is calculated at an equivalent biologic endpoint (e.g. time to reach a specific tumor size). This also allows for comparison of experimental manipulations that may change baseline tumor growth rates. 600mm³ is roughly the time to tumor tripling (as our tumors started at an approximate size of 200mm³). Tumor doubling or tumor tripling is a published and accepted measure of tumor growth delay/radiation response modulation. To clarify, we have better described this and cited relevant supporting literature (including a recent Nature Communications publication [Ref 65]) for this approach in the appropriate methods section [Page 27 and Paragraph 1; References 68].

- For the FaDu cell line all three clones are analyzed by WB, for the Detroit cell lines, only one clone is shown. Also, it is unclear which of the clones is used for which experiment.

Thank you for this comment. In response we have modified Figure 1D to include multiple gRNA Detroit562 populations. As shown in Suppl Figure 1D-E similar effects on survival were achieved with all of the isogenic populations generated using unique gRNAs. We have modified the manuscript [methods section Page 24 and Paragraph 2] to identify which cells were utilized for respective experiments.

Mitotic catastrophe is only scored on DNA staining of fixed cells. It is unclear if these cells have undergone mitosis. This should either be done by live-cell microscopy, or include relevant markers (combined phospho-HistoneH3 with cleaved caspase3 for instance). Binucleated cells are typically not considered as mitotic catastrophe.

Thank you for this critique which was also raised by Reviewer 1. Review of the literature has demonstrated to us that there are indeed varying definitions for mitotic catastrophe. The definition we used, founded upon quantification of multi-nucleation, is accepted by many groups (and was recently used for a paper published in Nature Communications [29]). Our goal in pursuing this series of experiments was to show that the DNA damage we observed with gH2AX and COMET assays is also consistent with changes seen at the cellular level. It does not appear that the reviewers are challenging the data about multinucleation, rather they question whether this is definitively mitotic catastrophe. In either case, the data strongly supports our observations that STING loss leads to less DNA damage and reduced appearance of multi-nucleate cells. To respond to the reviewers' critiques we have made the following changes 1) we state that multi-nucleation is a marker of mitotic catastrophe 2) we remove mention of mitotic catastrophe from the abstract, 3) we remove mention of mitotic catastrophe from the discussion, and 4) from the summation figure (Fig. 6h). We do keep this term as a label in Figure 3g for simplicity and believe we have provided enough information for the reader to accurately understand and interpret the data.

- The Sting re-expression experiments; include the parental cell line, to see if the re-expression is near endogenous or over-expressed.

Thank you for this suggestion. We have now modified the text and added a western blot comparing STING re-expression to WT cells showing over-expression of STING (Suppl Figure 1f). The data demonstrate that protein expression was indeed achieved in these experiments and that the effects of STING loss on cell survival are indeed due to on target effects of gene knockout given reversal of this effect with re-expression of STING.

- The reporting is not clear at multiple cases. The H2AX foci are shown as H2AX-positive cells versus negative cells. This is not acceptable, and should be reported as numbers of foci per cell, with individual cells as datapoints.

We agree with the reviewer that multiple methods have been used to describe DNA damage induced foci (e.g. Rad51, H2AX, etc). However, a commonly utilized, accepted and published methodology in the DNA damage/repair field is to report the number of H2AX foci positive cells. We have cited several studies that

have utilized this approach in the appropriate methods section [Page 25, Paragraph 2; Refs 51 and 63-65] to support our analysis.

- RNAseq is not described in the methods. Data is not deposited, unclear with the number of replicates is. Analysis is very limited. What is the effect of IR in these cells at the RNA levels? Also, it appears very arbitrary how ISG15 is selected as a DEG.

We apologize that the RNA-sequencing methodology was not included in the main methods section, but it was rather included in the supplemental materials section [Page 47-48] where it could be described in detail. We apologize that the biological replicate data were not explicitly reported. This has been remedied (n=3 replicates) in the supplemental materials sections [Page 48 paragraph 1]. All data have been deposited in GEO as described above. We have added a new supplementary table with the fold changes for all genes included in the heatmap in Figure 4B (Supplemental Table 3). Lastly, ISG15 was chosen as a representative DEG due to its role in the IFN response and known role in regulating ROS (both pathways significantly affected by STING loss) and its degree of change with STING loss.

- Page 3: CRISPR screens are common nowadays. The advantage over RNAi does not need to be introduced extensively.

We agree with the reviewer that the use of CRISPR-based techniques for genetic screening has become a valuable tool in biology. However, RNAi techniques continue to be heavily utilized. Of particular relevance is the continued use of RNAi based techniques in the field of DNA repair/radiobiology, as RNAi may have significant unintended consequences as indicated by the Elledge manuscript [Ref 12] due to off-target effects on critical DNA repair proteins such as Rad51. The use of RNAi in the field of radiation oncology is pervasive in abstracts, submitted manuscripts, and grant applications and frankly the use of this technique needs to be reconsidered. We hope our work will significantly advance the field of cancer biology and radiobiology as it provides a roadmap for identifying targets that combine favorably with radiation or other DNA damaging agents. As such we feel that it is worth introducing and reminding the reader of the advantages to CRISPR screens over RNAi in the context of DNA repair/DNA damage studies.

- Line 76: STING is not a DNA sensor itself, but a downstream target of DNA sensors.

Thank you for pointing this out, we have modified the sentence to clarify [Page 2, paragraph 1].

- Line 396: replicated -> replication

Thank you, this has been corrected in the manuscript.

Reviewer #3 (Remarks to the Author):

Summary

The manuscript describes the discovery that STING is a regulator of radiosensitivity in cancer cells as a result of a whole genome CRISPR screen. Surprisingly, this was the only gene discovered using this tool, but STING lies at a very interesting intersection of DNA sensing and immune regulation. This is a novel discovery and has the potential to be impactful.

However, while the paper identifies STING as a regulated gene, it does not adequately prove mechanism. Instead it transitions to the use of STING ligands as a cancer therapeutic. This has been demonstrated in a few different manuscripts, so is of lesser value. However, the major issue in this transition is that exogenous addition of STING does not fit with the initially described mechanism of STING as part of the DNA damage response in the cancer cell. The paper would be more valuable if it were able to prove how STING impacts radiosensitivity.

The manuscript is well-written and the experiments seem appropriate for their conclusions. There are no significant statistical issues. Figures are clear and well presented.

Major issues

The studies stop short of providing mechanism. While loss of STING decreases radiosensitivity, and results in a loss of IFN pathway activation and ROS production, we do not know whether these are mechanistic. Since only STING was identified by the genetic screen, and not TBK1, IFN receptors, or ROS-related genes, it is difficult to interpret relevance of the fact that these pathways are activated. These data would suggest that these pathways are not critical to the effects of STING loss. Additional experiments identifying whether the IFN pathway or the ROS-generating genes are necessary or sufficient for the observed effects would be important to make this case.

Thank you for this comment. The screen's identification of STING but not other upstream or downstream genes was at first puzzling to us as well. However, STING integrates multiple upstream signaling pathways and affects downstream expression for numerous genes. Our data are consistent with STING's coordination of a transcriptional response for regulating ROS and subsequent DNA DSB formation that is dependent upon many different transcriptional targets (as evidenced by RNAseq data). As such it would stand to reason that knockout of a single upstream or downstream effector of STING would not be sufficient to recapitulate the effects of STING loss on DNA damage. Rather our data are consistent with a critical role for STING as a nexus for controlling ROS homeostasis and we use ISG15 and GPX as mechanistic examples of how ROS is regulated.

With respect to further mechanistic experiments to test our model, we have followed the suggestion of Reviewer 2 to perform a series of new experiments using addition of NAC (a ROS scavenger) or H₂O₂ (ROS) in the setting of STING knockout. We found these results to be extremely informative. First, we show that measurements of ROS after the addition of H₂O₂ to WT or STING KO cells demonstrates a significantly reduced amount of ROS in STING KO (Fig. 4i). This shows that in the absence of DNA damaging agents, STING KO cells have an increased ability to buffer exogenous ROS. This finding is entirely consistent with data that shows enhanced GPX activity (Figs. 4g,h). Second, we demonstrate that the addition of H₂O₂ alone is sufficient to increase DNA damage as measured by γH2AX positive cells, but that H₂O₂ causes less DNA damage in the setting of STING KO (Fig. 4j), and is consistent with increased ROS buffering (Fig. 4i) and increased GPX activity (Figs. 4g,h). In the setting of radiation exposure, we use NAC to reduce radiation-induced ROS to levels below those seen in STING KO (Fig. 4i). This manipulation also significantly reduces radiation induced DNA damage (Fig. 4j). However, free radical scavenging with NAC is unable to further reduce radiation-induced DNA damage in cells with STING KO (Fig. 4j) which have lower levels of radiation-induced ROS (Fig. 4i). We also examined the addition of H₂O₂ in combination with radiation exposure and found that there were no significant enhancements of DNA damage in WT cells (Fig. 4j). However, the addition of H₂O₂ significantly enhanced DNA damage in STING KO cells, effectively 'rescuing' the induction of DNA damage by radiation. This line of experimentation provides significant insights into the effects of STING KO on ROS buffering and homeostasis, demonstrates that two separate approaches for enhancing ROS (H₂O₂ treatment or radiation exposure) both induce more DNA damage in WT vs STING KO cells, and that ROS scavenging or addition of H₂O₂ can duplicate or rescue the effects of STING KO, respectively. We believe this series of experiments provides new mechanistic data that strongly supports our findings regarding the role of STING in regulating ROS homeostasis.

The experiments using systemic STING ligands are difficult to interpret in view of the intrinsic DNA damage response model that is explored in the early part of the paper. While STING ligands can activate adaptive immunity in cancer models, the literature available thus far suggests that they work via STING expressed in stromal cells – in particular macrophages and dendritic cells, but occasionally endothelial cells. This is distinct from the effects of STING activation in the cancer cells by radiation-driven production of endogenous cGAS-generated cyclic dinucleotides. The authors make the case that STING expression in cancer cells is important to the response to radiation therapy, but they do not test STING treatment where the cancer cells or the stroma lack STING (either using STING^{-/-} mice or

their STING-/- cancer cells). It is plausible that the STING ligands are working through mechanisms that are independent of cancer cell expression of STING, as has been shown in other preclinical studies. Without requiring too many additional experiments or new mouse strains, it would be important to know whether RT+IV STING therapy remains effective if the cancer cells no longer express STING.

The reviewer brings up several valid points. First it has previously been shown that the effects of STING agonists are dependent upon the adaptive immune response, particularly the adaptive immune response through dendritic cells, macrophages, and endothelial cells ultimately by enhancing the CD8 positive T-cell response as is evidenced by experiments with T-cell depletion rescuing the effects of the STING agonist (e.g. Refs 20, 40 and Yang et al JCI 2019). In an attempt to minimize contributions of the adaptive CD8+ immune response we utilized athymic nude mice (functionally devoid of a CD8 positive T0-cell response) with STING WT or KO tumor cells, showing that even in the absence of the CD8 positive T-cell response tumor STING expression can modulate response to DNA damage.

In line with the reviewer's suggestion we have performed an additional animal experiment in a syngeneic HNSCC model using MOC1 tumor cells in C57BL/6J mice. MOC1 cells express negligible amounts of STING when compared to FaDu cells (Fig. 6f). We therefore tested the effects of STING agonist efficacy using an experimental design identical to that used for FaDu and Detroit562. In this syngeneic model both radiation and the STING agonist had modest anti-tumor effects alone, but as our in vitro data predict, the STING agonist did not significantly enhance radiation in this model (Fig 6g). Together these data provide additional support for our new understanding that tumor STING expression is an important component for the combined efficacy of a STING agonist and radiation when using fractionated radiation regimens.

The clinical data of patient outcome based on STING expression is very interesting, but it is notable that both HPV+ and HPV- patients are part of the study. It would be important to know whether these cohorts are differently represented in the STINGhi or STINGlo tumor/stromal groups, since others have shown that HPV+ cancers tend to express more STING in cancer cells (PMID: 29135982), and HPV+ patients generally have more favorable outcomes.

We thank the reviewer for this critique which raises similar concerns to that of Reviewer 2. Here we admit that we should not have used the term 'prognostic' as that claim would require further analysis. This description derailed the logic and goal of analyzing and presenting this patient data. The goal of this figure is to show that our insights into the biology of STING- that low tumor expression reduces DNA damage and improves survival after treatment with DNA damaging agents- correlates well with patient outcomes. In this figure we advance this concept further by demonstrating that both tumor and stromal STING expression levels can be used to identify patient subsets that have better (or worse) outcomes.

With respect to biomarkers in HNSCC, validation has been a major challenge to the field. Although we now accept p16/HPV as prognostic, many early studies analyzing large numbers of patients were confounded by clinical prognostic factors such as age or tumor stage. In fact, validation of p16/HPV as a prognostic factor reported by Ang et. al (NEJM 2010) required analysis of a 720 patient randomized trial! The analysis of a clinical trial and establishment of STING as an independent prognostic factor is beyond the scope of this manuscript. We are, however, in collaboration with other groups to analyze their institution's HNSCC TMA and have also initiated parallel projects to perform secondary analysis of completed clinical trials. These are independent studies being led by David Rimm (a co-author and pathologist) and have required significant expansion of contributors and lead authors for these future works.

To respond to the reviewers' critiques, we now provide additional data to support the association of STING expression with patient outcome. The impetus for performing the TMA analysis of STING protein expression was based on our observation that low STING RNA expression levels are associated with worse survival in the TCGA dataset. We did not include this data in the first version of the manuscript because the protein data is more directly relevant to the issue of tumor cell STING expression, but now see that we should have included this secondary dataset as an independent patient cohort. The TCGA data is now added

as Figure 5A (and below). Due to the size of this cohort (n=501 patients) we were able to analyze outcomes with respect to overall survival and the data shows that patients with low STING levels had a significantly worse overall survival. The low number of samples from our TMA (n=52) limited further analysis with respect to competing prognostic factors or the longer term endpoint of overall survival. When we revisited the TCGA dataset, we were surprised to learn even in this very large data set, the numbers of patients with known HPV status (n=103) were insufficient to perform an analysis of STING expression in HPV+ or HPV- patients (below). Thus, further analysis of larger patient cohorts is warranted and ongoing.

In summary, to address the reviewers' comments we have removed the term 'prognostic' and instead state that analysis of STING in larger cohorts is warranted, with discussion of limitations [page 17]. To provide confidence that this association is real, we now also show data from an independent patient cohort that demonstrates worse overall survival in patients with low STING expression.

In the revised manuscript we now further emphasize the importance of this clinical data with respect to our novel findings on STING's role in regulating ROS homeostasis. Figures 1-4 show that low STING expression alters an ROS transcriptional program, reduces ROS, reduces DNA damage, and enhances tumor cell survival. This data predicts that tumor cell STING levels are clinically important for sensitivity or resistance to DNA damaging agents. Figure 5a and 5b show that in HNSCC patients (often treated with DNA damaging agents), low STING at both the RNA and protein levels is associated with worse patient outcomes. Moreover, the clinical data from TMAs can be analyzed at a more granular level and supports our findings that tumor cell STING levels (and not just immune cell levels) are important, as low STING in either tumor or stromal compartments is associated with worse PFS (Fig 5e) and outperforms integrated analysis of STING protein across the tumor sample (Fig. 5b).

REVIEWERS' COMMENTS

Reviewer #1 (Remarks to the Author):

The authors have addressed this reviewer's concerns, so I consider the paper suitable for publication. The mechanism of STING impact on ROS is now made clearer with the new experiments included in Figure 4, showing the increase ability of STING KO cells to buffer exogenously added ROS in the absence of IR. In addition, the CRISPR and RNAseq data have now been made publicly available and the revised manuscript also strengthened the clinical relevance of the results.

Reviewer #2 (Remarks to the Author):

the authors put in great efforts to textually counter the many substantial flaws of the paper, data representation and data interpretation. although some minor issues were resolved, the manuscript still has most of my comments not addressed adequately. I feel that this study still does not provide the mechanistic detail nor the relevance to warrant publication.

Reviewer #3 (Remarks to the Author):

The authors have been responsive to review, have clarified the main message of the manuscript, and refined the mechanism.

This reviewer's major concerns related to:

1. The fact that STING was identified, but not the downstream genes that are proposed to mediate the effects, including IFN and ROS-related genes. The investigators have given strong responses, and clarified these in the manuscript.
2. The fact that exogenous STING ligands are more dependent on stromal STING expression than cancer cell STING expression. The authors provide arguments and some supportive data that cancer cell STING expression is important in one model.
3. The potential impact of varying proportions of HPV+ and HPV- tumors being in the survival analysis and impacting the analyses. The authors demonstrate in reviewer data that STING expression does not associate with outcome when considering either HPV+ or HPV- tumors alone, and suggest that the inability to measure different outcomes by subgroup analysis is a limitation in the number of patients available to analyze. While there remains a risk that the STING+ population is overrepresented with HPV+ patients, the language has been tempered and the data is suitable for publication.

For these reasons, this reviewer has no ongoing significant issues.

REVIEWERS' COMMENTS

Reviewer #1 (Remarks to the Author):

The authors have addressed this reviewer's concerns, so I consider the paper suitable for publication. The mechanism of STING impact on ROS is now made clearer with the new experiments included in Figure 4, showing the increase ability of STING KO cells to buffer exogenously added ROS in the absence of IR. In addition, the CRISPR and RNAseq data have now been made publicly available and the revised manuscript also strengthened the clinical relevance of the results.

Author Response: We thank the reviewer for their thoughtful suggestions and time to review this manuscript that have ultimately improved the presentation and quality of the manuscript. .

Reviewer #2 (Remarks to the Author):

the authors put in great efforts to textually counter the many substantial flaws of the paper, data representation and data interpretation. although some minor issues were resolved, the manuscript still has most of my comments not addressed adequately. I feel that this study still does not provide the mechanistic detail nor the relevance to warrant publication.

Author Response: We thank the reviewer for their time and consideration of our manuscript. Specifically we thank them for their suggestion of additional mechanistic studies related to ROS scavenging or generation as we believe these experiments significantly improved the main message of the manuscript and enhanced our mechanistic understanding of STING's involvement in the response to DNA-damaging therapies

Reviewer #3 (Remarks to the Author):

The authors have been responsive to review, have clarified the main message of the manuscript, and refined the mechanism.

This reviewer's major concerns related to:

- 1. The fact that STING was identified, but not the downstream genes that are proposed to mediate the effects, including IFN and ROS-related genes. The investigators have given strong responses, and clarified these in the manuscript.*
- 2. The fact that exogenous STING ligands are more dependent on stromal STING expression than cancer cell STING expression. The authors provide arguments and some supportive data that cancer cell STING expression is important in one model.*
- 3. The potential impact of varying proportions of HPV+ and HPV- tumors being in the survival analysis and impacting the analyses. The authors demonstrate in reviewer data that STING expression does not associate with outcome when considering either HPV+ or HPV- tumors alone, and suggest that the inability to measure different outcomes by subgroup analysis is a limitation in the number of patients available to analyze. While there remains a risk that the STING+ population is overrepresented with HPV+ patients, the language has been tempered*

and the data is suitable for publication.

For these reasons, this reviewer has no ongoing significant issues.

Author Response: We thank the reviewer for their careful review of our manuscript and their suggestions that have led to improvement of the manuscript.